# H3 ubiquitination by NEDD4 regulates H3 acetylation and tumorigenesis

Xian Zhang[1,2,3], Binkui Li[2], Abdol Hossein Rezaeian[2], Xiaohong Xu[2], Ping-Chieh Chou[1,2], Guoxiang Jin[1,2], Fei Han[1,2,3], Bo-Syong Pan[1,2], Chi-Yun Wang[1,2], Jie Long[1,2], Anmei Zhang[1,2], Chih-Yang Huang[4,5], Fuu-Jen Tsai[6,7], Chang-Hai Tsai[5,8], Christopher Logothetis[9] & Hui-Kuan Lin[1,2,3,4,5]

Dynamic changes in histone modifications under various physiological cues play important roles in gene transcription and cancer. Identification of new histone marks critical for cancer development is of particular importance. Here we show that, in a glucose-dependent manner, E3 ubiquitin ligase NEDD4 ubiquitinates histone H3 on lysine 23/36/37 residues, which specifically recruits histone acetyltransferase GCN5 for subsequent H3 acetylation. Genome-wide analysis of chromatin immunoprecipitation followed by sequencing reveals that NEDD4 regulates glucose-induced H3 K9 acetylation at transcription starting site and enhancer regions. Integrative analysis of ChIP-seq and microarray data sets also reveals a consistent role of NEDD4 in transcription activation and H3 K9 acetylation in response to glucose. Functionally, we show that NEDD4-mediated H3 ubiquitination, by transcriptionally activating IL1α, IL1β and GCLM, is important for tumour sphere formation. Together, our study reveals the mechanism for glucose-induced transcriptome reprograming and epigenetic regulation in cancer by inducing NEDD4-dependent H3 ubiquitination.

[1] Department of Cancer Biology, Wake Forest School of Medicine, Medical Center Boulevard, Winston-Salem, North Carolina 27157, USA. [2] Department of Molecular and Cellular Oncology, The University of Texas MD Anderson Cancer Center, Houston, Texas 77030, USA. [3] The University of Texas Graduate School of Biomedical Sciences at Houston, Houston, Texas 77030, USA. [4] Graduate Institute of Basic Medical Science, China Medical University, Taichung 404, Taiwan. [5] Department of Biotechnology, Asia University, Taichung 41354, Taiwan. [6] College of Chinese Medicine, China Medical University, Taichung 40402, Taiwan. [7] Department of Medical Genetics, Pediatrics and Medical Research, China Medical University Hospital, Taichung 40402, Taiwan. [8] Center of Molecular Medicine, China Medical University Hospital, Taichung 40402, Taiwan. [9] Department of Genitourinary Medical Oncology, University of Texas MD Anderson Cancer Center, Houston, Texas 77030, USA. Correspondence and requests for materials should be addressed to H.-K.L. (email: hulin@wakehealth.edu).

Histone acetylation is one of the well-known histone modification that is frequently decorated on histone H3 and H4, which are often markers for the open up of chromatin and activation of gene transcription[1,2]. Histone acetylation, in essence, is an outcome of transferring an acetyl group from acetyl-CoA to the lysine residue on histone protein by specific enzymes known as histone acetyltransferases (HATs) that often resides in larger chromatin remodelling complexes. Several families of HATs, including Gcn5-related N-acetyltransferases (GNATs), MYST HATs and others, have been discovered and are responsible for histone acetylation[3,4]. Each HAT catalyses histone acetylation at their own preferred lysine sites and is involved in distinct biological processes. Thus, it is important to study how those HATs are selectively utilized under specific conditions and what are the underlying mechanisms to convey the specificity.

Eukaryotic cells rapidly adapt to the changes in its environment to survive and proliferate. One way of adaptation is achieved by a precise reprogramming of gene expression to fine-tune the cellular functions in the face of environmental changes. The dynamics of histone modification is critically responsible for the altered gene transcription. With regard to stress-induced changes in histone modification, there are basically two aspects. Stress-sensing machinery in the cell transduces environmental changes to cellular signalling and activated signalling cascade eventually triggers the histone modification machinery to regulate gene transcription[5]. Alternatively, environmental changes such as metabolism-related changes have been shown to affect histone modification directly, as many metabolic intermediates are co-factors or substrates for the enzymatic reaction of histone modification[6]. Glucose feeding, which causes a rapid increase in histone acetylation, is proposed to achieve this through acetyl-CoA generation by either direct conversion of pyruvate to acetyl-CoA in the nucleus or citrate to acetyl-CoA in the cytoplasm[7–9]. As HAT critically links acetyl-CoA to histone acetylation, it is important to understand whether and how glucose utilizes specific HATs for histone acetylation. Gcn5 is reported to be involved in the glucose-induced histone acetylation in yeast[10], but the underlying mechanism of how glucose determines HAT for the glucose-induced acetylation of histone is not well understood and needs to be further studied.

Ubiquitination is a one of the well-studied posttranslational modifications that is critically involved in many cellular processes. Ubiquitination modification ranges from mono, poly or even multi-mono, multi-polyubiquitination, which decide the distinct fate of the proteins, including proteasomal degradation, lysosomal degradation or protein trafficking[11]. Histones such as histone H2A and H2B are well known to be ubiquitinated and their ubiquitination cross-talks with other histone modifications and regulates gene transcription[12]. Ubiquitination is triggered by the cooperation of three enzymes, including E1 ubiquitin-activating enzyme, E2 ubiquitin-conjugating enzyme and E3 ubiquitin ligase, the one that determines the substrate specificity of the reaction. NEDD4 is an HECT domain ubiquitin E3 ligase that triggers ubiquitination on various proteins, often leading to proteasomal degradation of its substrates or receptor endocytosis[13]. NEDD4 deficiency causes embryonic lethality and profound decrease in insulin growth factor-1 and insulin signalling in multiple cell types[13–15]. NEDD4 contains a C2 domain on its amino terminal and a HECT domain on its carboxy terminal, which interact with each other, leading to a self-inhibitory formation. Cellular $Ca^{2+}$, which binds to NEDD4 C2 domain, or phosphorylation at both domains, could disrupt the self-inhibitory structure and activate NEDD4 E3 ligase activity[16,17].

In this study, we discover a histone H3 ubiquitination mark, which is critical in transcriptional regulation. We showed H3 ubiquitination is physiologically induced by glucose and is essential for H3 acetylation at K9, K14, K27 and K56 by selective recruitment of HAT GCN5. Genome-wide analysis of chromatin immunoprecipitation followed by sequencing (ChIP-seq) data sets revealed that glucose induces H3 acetylation at transcription start site (TSS) in a gene-specific manner (around 2,000 genes) and ~40% of these acetylation events are regulated by NEDD4. Integrative analysis combining ChIP-seq and gene expression microarray data sets further revealed that these acetylation events at TSS are significantly correlated with the expression of their target genes, many of which are involved in cancer-related pathways, in a NEDD4-dependent manner. Interestingly, we also found that H3 ubiquitination, together with NEDD4, GCN5 and histone H3.3, are novel regulators for tumour sphere formation. Target genes of glucose-induced H3 ubiquitination, such as interleukin (IL)-1α, IL1β and glutamate-cysteine ligase regulatory subunit (GCLM), are also important factors for tumour sphere formation.

## Results

**NEDD4 ubiquitinates H3 upon glucose stimulation.** As depleting subunits recognizing known motifs or histone modifications in HAT complex did not affect glucose-induced global histone H3 acetylation[18], we speculated that there are other previously unknown H3 modifications, which may be involved in the global recruitment of HAT. By analysing the published large-scale quantitative mass spectrometry data sets[19–21], we found that H3 proteins were ubiquitinated on multiple lysine (K) residues (Fig. 1a and Supplementary Fig. 1a). Although these proteomic studies are mostly carried out under non-stimulus conditions, it is critical to know which physiological cues can drive this ubiquitination modification and what functions it may play. To address these questions, we challenged cells with various physiological stimuli and performed *in vivo* ubiquitination assay to detect ubiquitination of endogenous H3 proteins. Of these stimuli (at 4 h), H3 ubiquitination was drastically inhibited by glucose deprivation, whereas it was also reduced by glutamine deprivation, serum starvation to a lesser extent and not affected by infrared and $H_2O_2$ treatment (Fig. 1b and Supplementary Fig. 1b). We also excluded the possibility that glucose starvation causes irreversible damages to the cells, thereby leading to the reduction of H3 ubiquitination indirectly, as add-back of glucose to cells under glucose deprivation readily recovered H3 ubiquitination (Fig. 1c and Supplementary Fig. 1c). Although H3 ubiquitination is regulated by the glucose status, we found that H3 ubiquitination is not affected by cell cycle (Supplementary Fig. 1d). Accordingly, these results suggest that glucose is a *bona fide* physiological activator for H3 ubiquitination.

We next determined which E3 ligase is responsible for glucose-induced H3 ubiquitination. By screening a panel of ubiquitin E3 ligases available in our laboratory, NEDD4 (also known as neural precursor cell expressed developmentally downregulated protein 4) was identified to be a potential E3 ligase for H3 ubiquitination, as wild-type (WT) NEDD4 could promote H3 ubiquitination, but not E3 ligase dead mutant NEDD4 (NEDD4-CS) (Fig. 1d,e and Supplementary Fig. 1e,f). In line with this notion, NEDD4 knockdown abolished H3 ubiquitination (Fig. 1f). To investigate whether NEDD4 is a direct E3 ligase for H3, we performed *in vitro* ubiquitination assay by mixing recombinant active form of NEDD4 with various E2 ubiquitin-conjugating enzymes and histone octamer. We found that NEDD4 in combination with UbcH7 effectively triggered *in vitro* H3 ubiquitination, although UbcH5 a/b/c mix or UbcH6 also exhibited curtain enzymatic activity (Fig. 1g). Next we investigated whether NEDD4 is required for glucose-induced H3 ubiquitination. By performing immunoprecipitation assay for

endogenous proteins, we found that NEDD4 knockdown or knockout abolished endogenous H3 ubiquitination induced by glucose treatment (Fig. 1h and Supplementary Fig. 1g,h), indicating a physiological role of NEDD4 in mediating glucose-induced H3 ubiquitination. We further asked the question whether the glucose levels could orchestrate E3 ligase activity of NEDD4. Our data showed that NEDD4 overexpression failed to

trigger H3 ubiquitination under glucose deprivation, whereas add-back of glucose readily rescued NEDD4 overexpression-induced H3 ubiquitination (Fig. 1i). As we have also noticed that H3 ubiquitination in some of our experiments displayed multiple ubiquitination bands at higher molecular weight, we sought to test whether H3 undergoes polyubiquitination or multi-monoubiquitination. We found that ubiquitin with its all

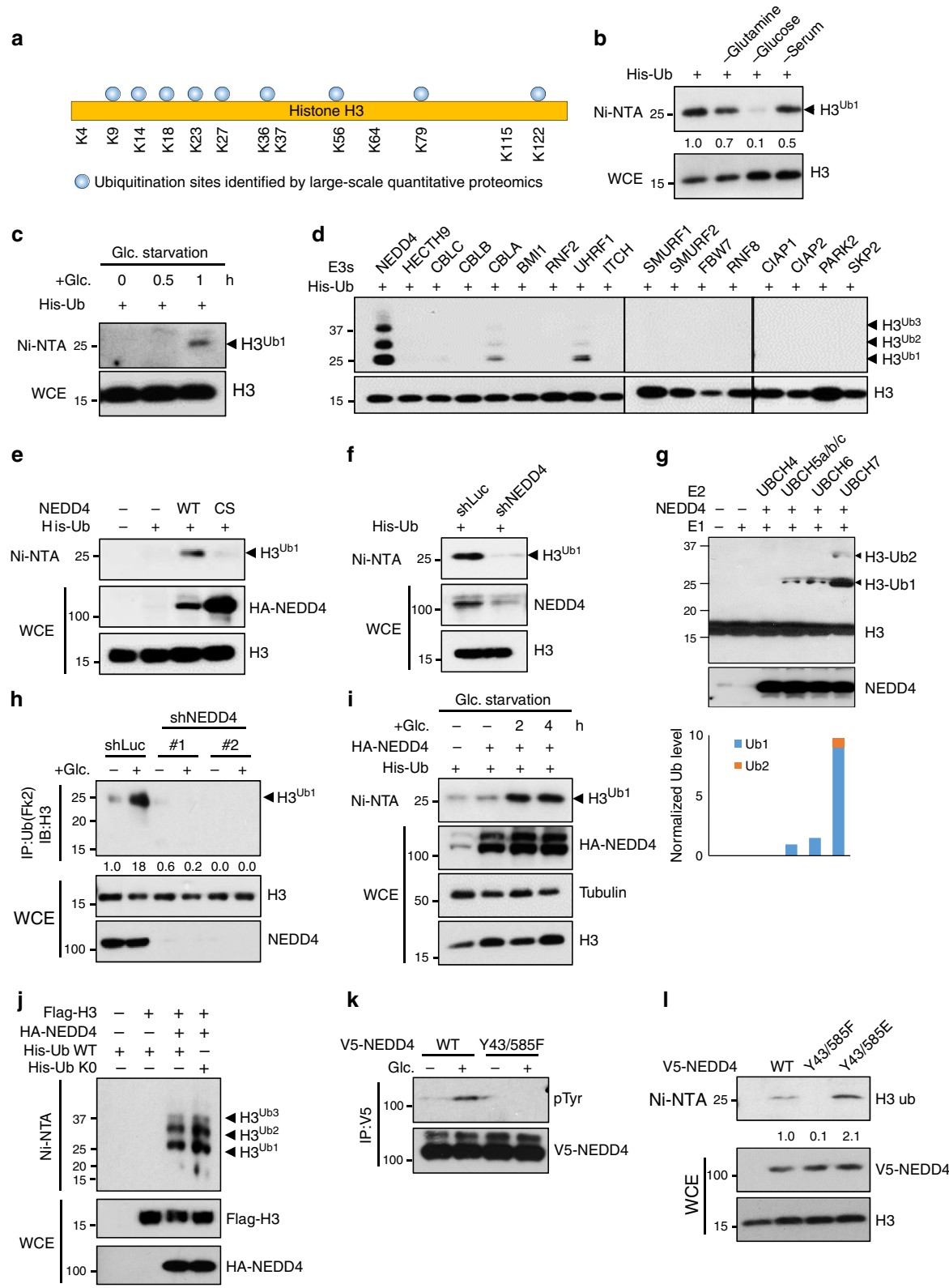

K mutated to R (Ub K0) did not affect H3 ubiquitination pattern in the *in vivo* ubiquitination assay (Fig. 1j), indicating that NEDD4 monoubiquitinates H3 at multiple sites. Moreover, although NEDD4 is previously shown to regulate protein mostly in the cytosol and plasma membrane, a few nuclear substrates for NEDD4 are found[22,23]. Consistently, we found that NEDD4 could be detected both in cytosol and nucleus by using biochemical fractionation and immunofluorescence assays (Supplementary Fig. 2a,b), suggesting that NEDD4 can target its protein substrates in the nucleus such as H3. Collectively, these data provide strong evidence that NEDD4 is a direct E3 ligase for H3 and is responsible for glucose-induced H3 ubiquitination.

To understand how glucose promoted NEDD4-mediated H3 ubiquitination, we first examined whether glucose regulates the subcellular localization of NEDD4. Biochemical fractionation assay revealed that NEDD4 localization in the cytoplasm, nucleoplasm and chromatin remained unchanged upon glucose treatment (Supplementary Fig. 2c). Alternatively, we speculated that glucose treatment may activate NEDD4 through altering the posttranslational modification status of NEDD4. As glucose is related to the cellular level of ATP and acetyl-CoA, which are important cofactors for protein phosphorylation and acetylation, we evaluated overall phosphorylation and acetylation of NEDD4 protein. Interestingly, we found that tyrosine phosphorylation of NEDD4, but not NEDD4 serine/threonine and acetylation, is strongly induced b yglucose treatment (Supplementary Fig. 2d). As a previous report has shown that NEDD4 phosphorylation at tyrosine (Y) 43 and Y585 promotes NEDD4 E3 ligase activity[16], we examined whether glucose also induced NEDD4 phosphorylation at tyrosine (Y) 43 and Y585, and found that mutating Y43/585 to phenylalanine (F) blocked glucose-induced NEDD4 tyrosine phosphorylation (Fig. 1k), indicating that glucose induces NEDD4 tyrosine phosphorylation at Y43 and Y585, which may then activate NEDD4 through Y43/585 phosphorylation. In support of this notion, NEDD4 Y43/585F mutant failed to trigger H3 ubiquitination, but phosphorylation mimetic mutant Y43/585E enhanced H3 ubiquitination more efficiently than WT NEDD4 (Fig. 1l). Importantly, Y43/585E mutant rescued H3 ubiquitination under glucose deprivation to the level similar to that triggered by WT NEDD4 in glucose-stimulated conditions (Supplementary Fig. 2e). Collectively, our results suggest that glucose induces NEDD4 activation by inducing NEDD4 tyrosine phosphorylation at Y43 and Y585, in turn leading to H3 ubiquitination.

**Glucose-induced H3 ubiquitination regulates H3 acetylation.** Next we determined whether NEDD4-mediated H3 ubiquitination regulates H3 acetylation. Similar to H3 ubiquitination, we found that glucose treatment readily induced H3 acetylation on K9, K14, K27 and K56 sites, but had no effect on H3 phosphorylation, H3 di- and trimethylation or H3 acetylation at other lysine sites (Fig. 2a and Supplementary Fig. 3a–e). Notably, NEDD4 knockdown or knockout specifically impaired glucose-induced H3 acetylation at glucose-responsive lysine residues without affecting acetyl-CoA levels (Fig. 2a and Supplementary Fig. 3f,g), suggesting that NEDD4 deficiency essentially mimics the effect of glucose on H3 acetylation and this is probably not through regulating the acetyl-CoA level. Although mammalian H3 contains three isoforms (H3.1, H3.2 and H3.3), H3.3 is found to be the major variant decorated by H3 acetylation[24]. Consistent with the previous findings, we demonstrated that H3.3 knockdown effectively reduced H3 acetylation on K4, K9, K14, K23, K27 and K56, but not K18 and K36 (Fig. 2b). Similar to NEDD4 knockdown and glucose stimulation, H3.3 knockdown also did not alter H3 methylation (Fig. 2a,b). Although NEDD4 knockdown does not reduce the protein levels of each H3 variant (Supplementary Fig. 3h), we found that H3 ubiquitination mainly occurs on H3.3 compared with H3.1/H3.2 (Supplementary Fig. 3i,j), suggesting that H3.3 is likely to be the major substrate for H3 ubiquitination. As H3.3 is also the major substrate for H3 acetylation, we then used H3.3 construct to determine NEDD4-dependent ubiquitination sites. After serial mutagenesis on all H3.3 lysine residues, we found that mutation of K23 or K36/37 effectively blocked glucose- and NEDD4-mediated H3.3 ubiquitination both *in vivo* and *in vitro* (Fig. 2c,d and Supplementary Fig. 3k–m), suggesting that NEDD4 specifically ubiquitinates H3 on K23, K36 and K37 residues. The existence of endogenous ubiquitination modification on these sites was also supported by previous large-scale quantitative mass spectrometry analysis[20,21] (Fig. 1a and Supplementary Fig. 1a). To determine the causal relationship between H3 ubiquitination and acetylation, we stably expressed WT H3.3 and H3.3 ubiquitination-deficient mutant in Hep3b cells to examine H3 acetylation. Consistent with the effect of glucose and NEDD4 on H3 acetylation, H3.3 K23/36/37R is defective in H3 K9/K14 acetylation (Fig. 2e,f and Supplementary Fig. 3n,o), whereas single or double mutation also displayed partial effect on H3 acetylation (Supplementary Fig. 3p). In addition, we performed kinetic study for H3 ubiquitination and K9 acetylation to further elucidate their regulation. Adding back glucose readily induced

**Figure 1 | NEDD4 ubiquitinates H3 upon glucose stimulation.** (**a**) Summary of H3 ubiquitination sites identified in various large-scale quantitative proteomics studies. (**b**) Glucose deprivation abolished H3 ubiquitination. 293T cells were transfected with his-ubiquitin plasmid (His-Ub) for 36 h and treated with various stresses for 4 h before *in vivo* ubiquitination assay to access the H3 ubiquitination (see experimental procedures for details). (**c**) Add-back of glucose recovered H3 ubiquitination. 293T cells were transfected with his-ubiquitin plasmid for 36 h, then glucose-starved for 4 h and added-back glucose for indicated times (see experimental procedures for detail) before *in vivo* ubiquitination assay. (**d**) Screening of E3 ligases for H3 ubiquitination. 293T cells were transfected with his-ubiquitin plasmid and various E3 ligases constructs for *in vivo* ubiquitination assay. (**e**) NEDD4 E3 ligase dead mutant (CS mutant) failed to trigger H3 ubiquitination. 293T cells were transfected with his-ubiquitin plasmid and WT NEDD4 or NEDD4 CS mutant construct for *in vivo* ubiquitination assay. (**f**) NEDD4 knockdown abolished H3 ubiquitination. Control and NEDD4 knockdown 293T cells were transfected with his-ubiquitin plasmid for *in vivo* ubiquitination assay. (**g**) NEDD4 ubiquitinated H3 *in vitro*. *In vitro* ubiquitination assay was performed for recombinant NEDD4 and histone octamer (see experimental procedures for details). Reaction products were then assessed by western blotting using anti H3 antibody. H3 mono- and di-ubiquitination have predicted molecular weights of ~25 kDa and ~33 kDa. S.E. and L.E. are abbreviations for shorter exposure time and longer exposure time, respectively. (**h**) NEDD4 knockdown abolished glucose-induced H3 ubiquitination. Hep3B cells were glucose starved for 4 h and added-back glucose for 2 h before immunoprecipitation assay for endogenous ubiquitinated proteins (see experimental procedures for details). H3 ubiquitination was then visualized by western blotting. (**i**) Add-back of glucose recovered NEDD4 overexpression induced H3 ubiquitination. 293T cells were transfected with his-ubiquitin and NEDD4 plasmids for 36 h, then glucose-starved for 4 h and added-back glucose for indicated times before *in vivo* ubiquitination assay. (**j**) NEDD4 triggered monoubiquitination on H3. 293T cells were transfected with Flag-H3, HA-NEDD4, His-Ub WT and His-Ub K0 as indicated before *in vivo* ubiquitination assay. (**k**) Glucose-induced NEDD4 phosphorylation at Y43 and Y585. 293T cells transfected with WT or Y43/585F NEDD4 plasmids were treated with glucose and harvested for IP. (**l**) NEDD4 phosphorylation is required for H3 ubiquitination. 293T cells transfected with WT, Y43585F or Y43/585E NEDD4 plasmids were harvested for *in vivo* ubiquitination assay.

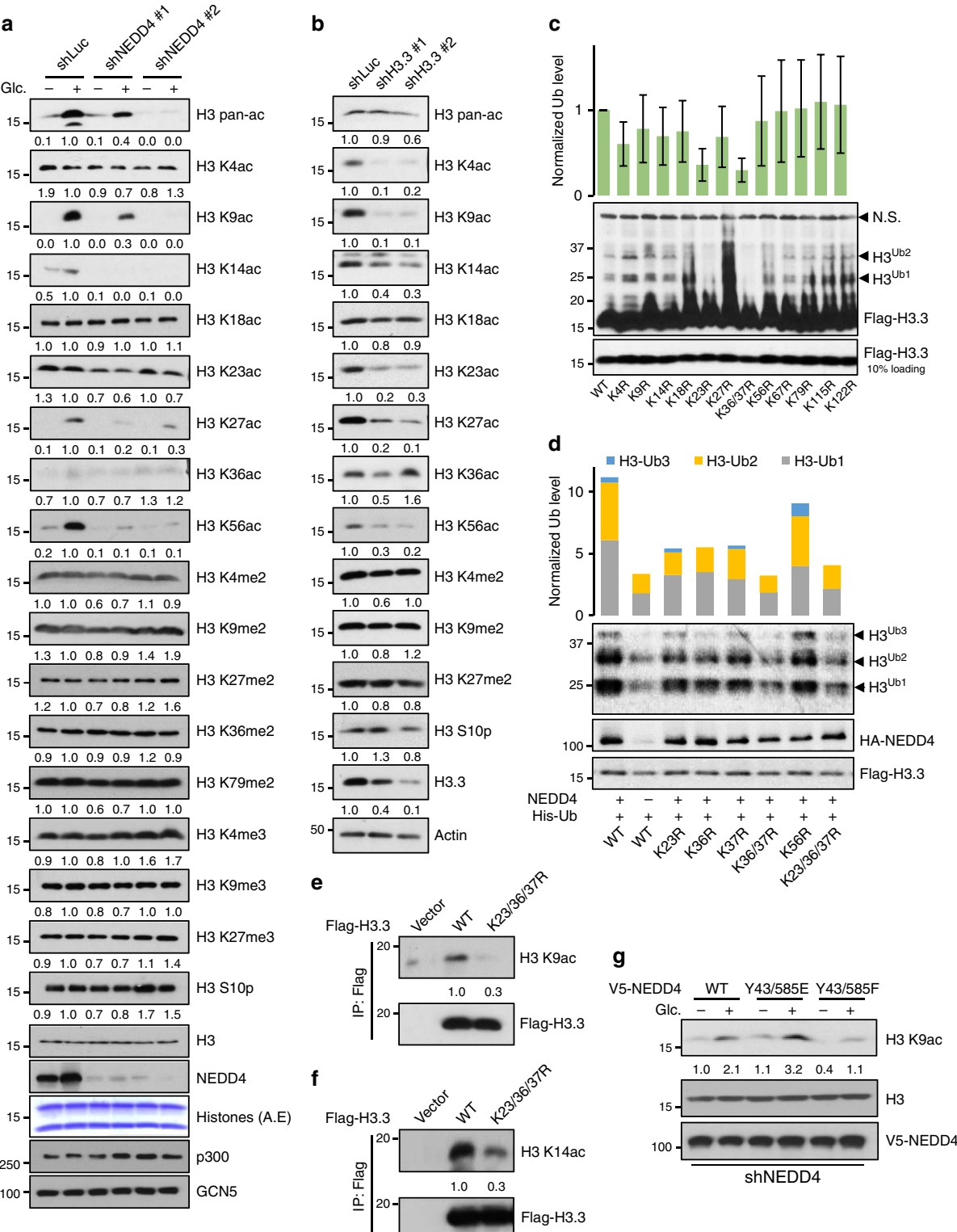

**Figure 2 | Glucose-induced H3 ubiquitination by NEDD4 is required for H3 K9/K14 acetylation.** (**a**) NEDD4 knockdown abolished glucose-induced H3 K9, K14, K27 and K56 acetylation. Control and NEDD4 knockdown Hep3B cells were glucose starved for 4 h and added-back glucose for 3 h before whole-cell extraction for western blot analysis (see experimental procedures for details). A.E., acid extraction. (**b**) H3.3 knockdown abolished H3 K9, K14, K27 and K56 acetylation. Control and H3.3 knockdown Hep3B cells were lysed for western blot analysis. (**c**) H3 K23R and K36/37R mutant abolished glucose-induced H3 ubiquitination. Hep3B cells expressing various Flag-H3.3 constructs were glucose-starved for 4 h and added-back glucose for 2 h before chromatin fractionation assay. Ubiquitination levels were normalized to input ($n = 5$, mean ± s.e.m.). NS, nonspecific band. (**d**) H3 K23 and K36/37R mutant abolished NEDD4 overexpression induced H3 ubiquitination. 293T cells were transfected with his-ubiquitin, NEDD4 plasmids and various Flag-H3.3 constructs for 36 h before *in vivo* ubiquitination assay. Ubiquitination levels were normalized to input. (**e,f**) H3 K23/36/37R is defect in H3 K9/K14 acetylation. WT or K23/36/37R Flag-H3.3 was stably expressed in Hep3B cells and immunoprecipitated for western blot analysis. (**g**) NEDD4 knockdown Hep3B cells transfected with WT, Y43/585E or Y43/585F mutant were treated with glucose and harvested for western blot analysis.

H3 ubiquitination in 1 h, whereas acetylation of H3 at K9 occurred around 2 h (Supplementary Fig. 3q), indicating H3 ubiquitination occurs earlier than H3 K9 acetylation and could be upstream of H3 acetylation. To test whether NEDD4 tyrosine phosphorylation at Y43 and Y583 plays a critical role in H3 acetylation, we restored NEDD4 WT, Y43/585F and Y43/585E mutant in the NEDD4 knockdown cells, to examine their effect on H3 acetylation. We found that NEDD4 Y43/585F mutant failed to rescue glucose-induced H3 K9 acetylation, but NEDD4 Y43/585E mutant fully rescued glucose-induced H3 K9 acetylation and executed such effect even more efficiently than NEDD4 WT (Fig. 2g), confirming that glucose activated NEDD4 through NEDD4 tyrosine phosphorylation. Taken together, our data suggest that glucose-induced H3 ubiquitination by NEDD4 selectively regulates H3 acetylation at specific lysine sites, including K9 and K14.

**NEDD4 regulates H3 acetylation at TSS and enhancers.** H3 K9 acetylation is known to localize at the TSS and generally regulates gene transcription in mammalian cells[25,26]. Having shown that glucose treatment increases total or cellular level of H3 K9 acetylation in a NEDD4-dependent manner (Fig. 2a), we further tested this regulation at chromatin level. Therefore, genome-wide pattern of H3 K9 acetylation was determined by ChIP-seq assay. Consistent with western blotting result in Fig. 2a, we found that glucose globally enhanced H3 K9 acetylation at TSS of genes and such effect was impaired upon NEDD4 knockdown by ChIP-seq assay (Fig. 3a and Supplementary Fig. 4a,b). Our results also indicated that glucose induces H3 K9 acetylation at TSS of around 2,000 genes and 40% of those acetylation events were NEDD4 dependent (Fig. 3b,c), highlighting the critical role of

NEDD4 in glucose-induced H3 K9 acetylation at TSS. Meta-analysis also revealed that glucose promoted H3 K9 acetylation on known enhancers and this event was also NEDD4 dependent (Fig. 3d and Supplementary Fig. 4c), suggesting that NEDD4 may also participate in the activation of enhancer in response to glucose stimulation. To further evaluate the role of H3 ubiquitination in the glucose-dependent transcriptional activation, we also performed ChIP-seq assay using anti-conjugated ubiquitin antibody (FK2). By analysing the pool of genes with glucose-inducible H3 K9ac at TSS, we found that, similar to H3 K9ac, the occupancy of ubiquitinated proteins was also enriched at TSS of those genes (Supplementary Fig. 4d). Meanwhile, the occupancy of ubiquitinated proteins at TSS was reduced in H3.3 K23/36/37R restored cells compared with H3.3 WT restored cells (Supplementary Fig. 4d), suggesting that this ubiquitination signal is partially related to H3.3 ubiquitination. Together, these data provide further evidence in support of NEDD4-mediated H3.3 ubiquitination is enriched at TSS and regulates glucose-induced H3 K9 acetylation.

**H3 ubiquitination regulates gene transcription.** We next determined whether glucose-induced H3 ubiquitination by NEDD4 regulates transcription and, if so, whether H3 K9 acetylation is associated with this regulation. To this end, genome-wide differential gene expression pattern was determined by gene expression microarray. Integrative analysis of microarray and ChIP-seq data sets revealed that NEDD4 knockdown caused downregulation of around 5,000 genes and 50% of these genes showed reduced H3 K9 acetylation at TSS upon NEDD4 knockdown (Fig. 4a,b). In addition, we applied gene set enrichment analysis (GSEA)[27,28] using modified Kolmogorov–Smirnov (K–S)

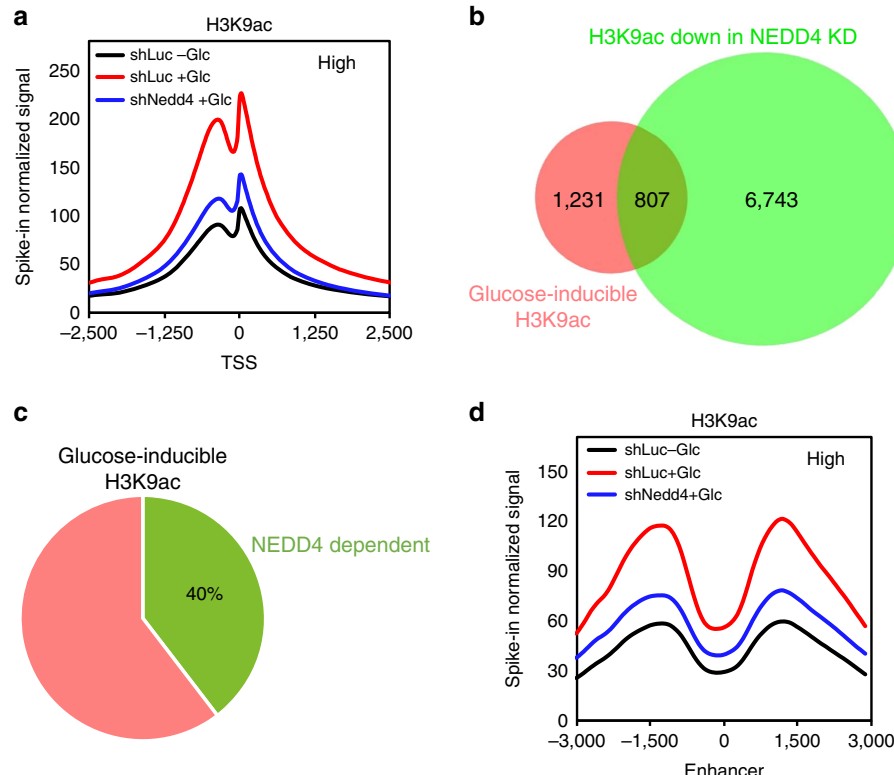

**Figure 3 | NEDD4 is required for glucose-induced H3 K9 acetylation at TSS and enhancers.** NEDD4 knockdown impaired glucose-induced genome-wide H3 K9 acetylation at TSS and enhancer regions. ChIP-seq was performed for control and NEDD4 knockdown Hep3B cells before or after adding-back of glucose for 3 h. Shown were global H3 K9ac profiles at TSS (**a**), Venn diagram of genes with differential H3 K9ac peaks at TSS under glucose treatment and NEDD4 knockdown condition (**b,c**), and global H3 K9ac profiles at known enhancers (**d**). See experimental procedures for details.

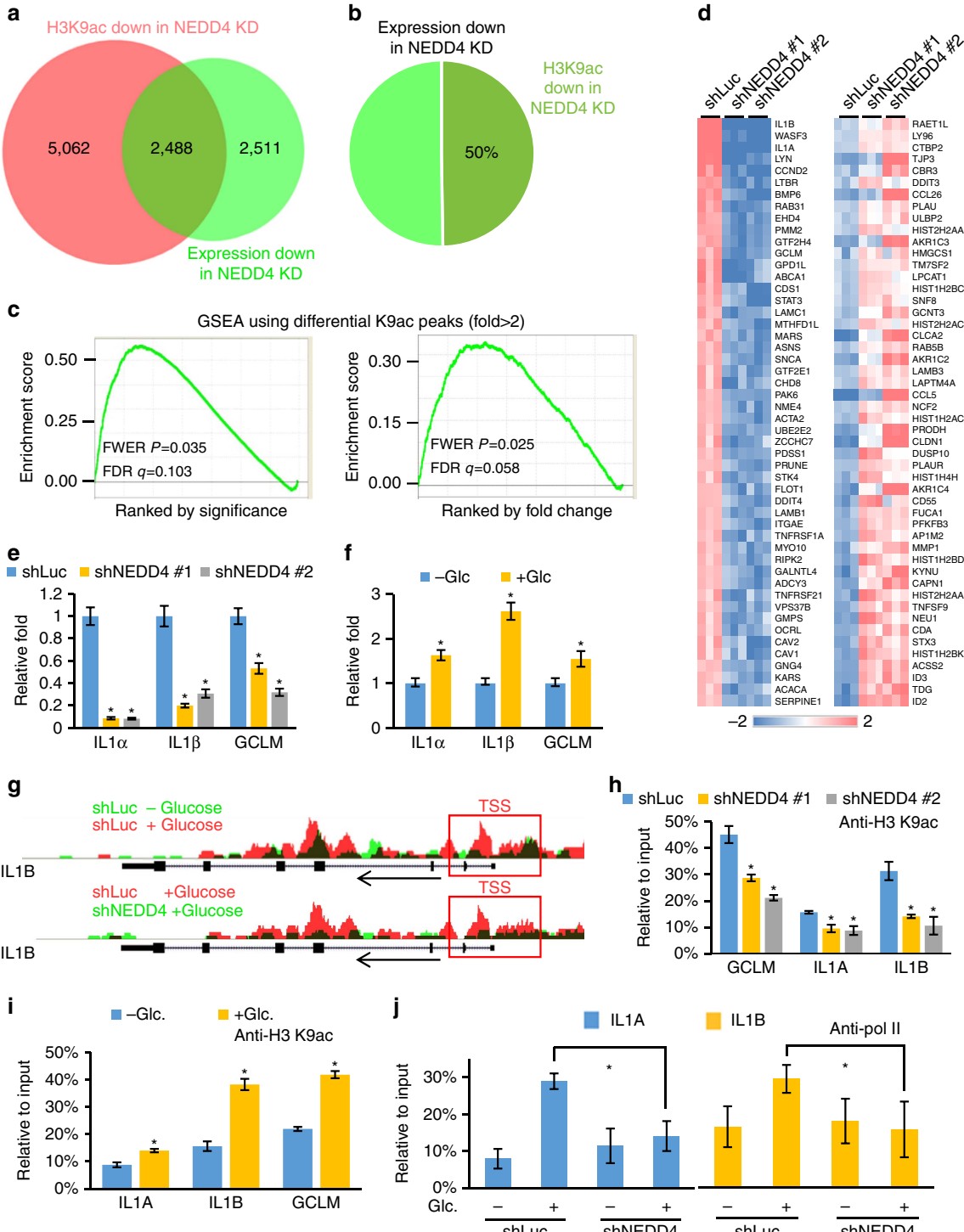

**Figure 4 | H3 ubiquitination regulates transcription and recruits GCN5 for H3 acetylation.** (**a–c**) NEDD4 regulates H3 K9ac at TSS of NEDD4 target genes. Shown were Venn diagram of genes with differential expression or differential H3 K9ac at TSS. GSEA was performed to evaluate the distribution of genes that show down-regulation of H3K9ac at TSS in NEDD4 knockdown cells in microarray-derived gene list, which is rank ordered either by T-test or fold change. (**d**) Heat map view of top and bottom gene list of microarray data sets. Microarray analysis for total RNA was performed for control and NEDD4 knockdown Hep3B cells. (**e**) NEDD4 knockdown impaired IL1α, IL1β and GCLM expression. qPCR was performed to analyse the mRNA level in control and NEDD4 knockdown Hep3B cells ($n = 3$, mean ± s.e.m.). (**f**) IL1α, IL1β and GCLM were induced by glucose. Hep3B cells were glucose starved for 4 h and added-back glucose for 6 h before qPCR analysis ($n = 3$, mean ± s.e.m.). (**g**) UCSC genome browser view of ChIP-seq H3 K9ac signals along the *IL1B* gene. (**h**) NEDD4 knockdown impaired H3 K9ac at TSS of IL1α IL1β and GCLM genes. ChIP-qPCR using anti-H3 K9ac antibody was performed for control and NEDD4 knockdown Hep3B cells ($n = 3$, mean ± s.e.m.). (**i**) H3 K9ac was induced at TSS of IL1 α IL1β and GCLM genes by glucose. Hep3B cells were glucose-starved for 4 h and added-back glucose for 6 h before ChIP-qPCR analysis using anti-H3 K9ac antibody ($n = 3$, mean ± s.e.m.). (**j**) NEDD4 knockdown impaired glucose-induced polymerase II (pol II) binding at TSS of *IL1A* and *IL1B* genes. Control and NEDD4 knockdown Hep3B cells were glucose-starved for 4 h and added-back glucose for 6 h before ChIP-qPCR analysis using anti-pol II antibody ($n = 3$, mean ± s.e.m.). All asterisks (*) represent $P < 0.05$, using Student's T-test.

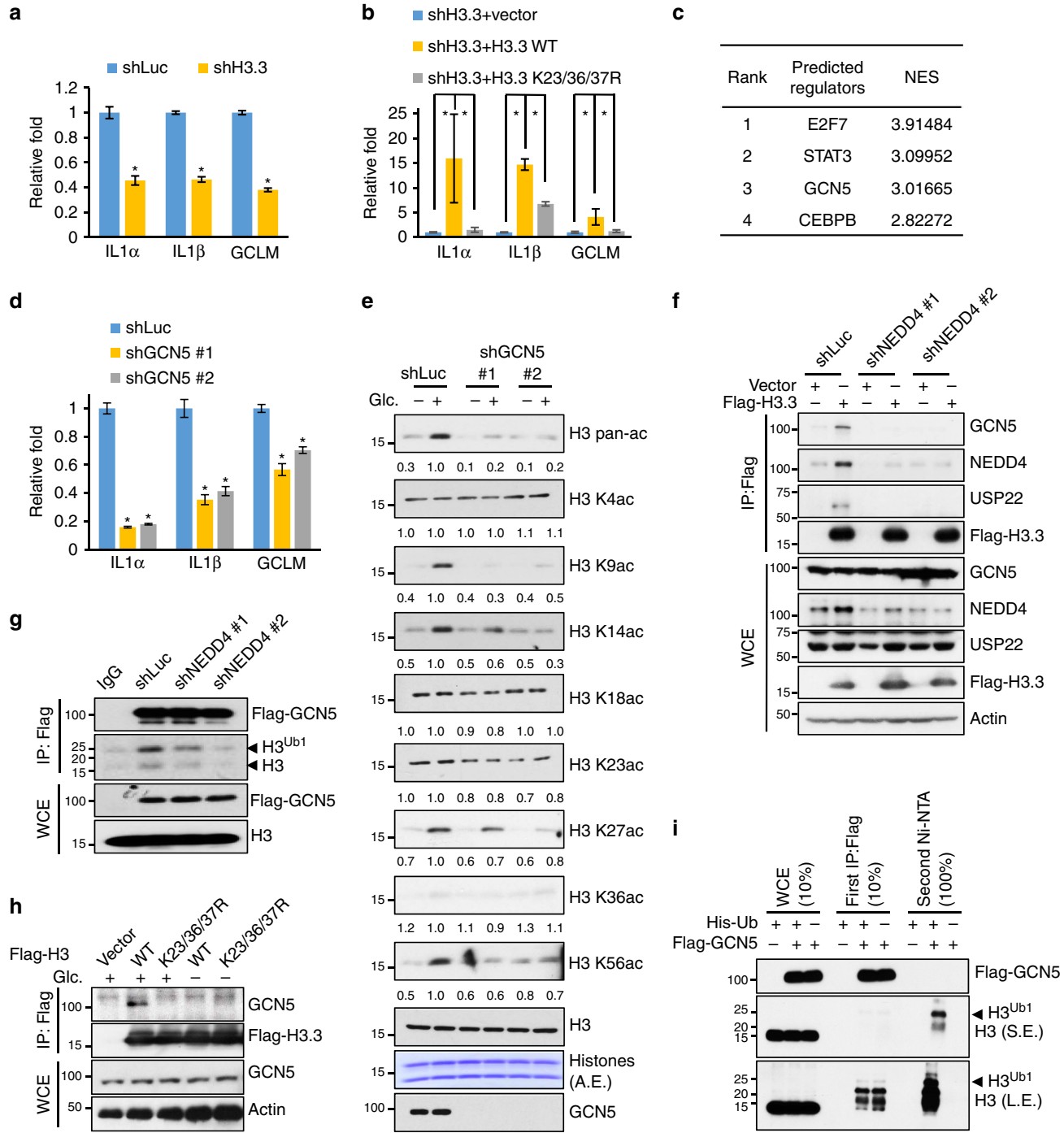

**Figure 5 | H3 ubiquitination specifically recruits GCN5 for H3 acetylation.** (**a**) H3.3 knockdown impaired IL1α, IL1β and GCLM expression. qPCR was performed to analyse the mRNA level in control and H3.3 knockdown Hep3B cells ($n = 3$, mean ± s.e.m.). (**b**) H3 ubiquitination deficiency impaired IL1α, IL1β and GCLM expression. qPCR was performed to analyse the mRNA level in vector, H3.3 WT and H3.3 K23/36/37R mutant restored H3.3 knockdown Hep3B cells ($n = 3$, mean ± s.e.m.). (**c**) GCN5 is predicted as one of the potential regulator for NEDD4 target genes. Top 5,000 upregulated genes from microarray data sets in Fig. 4d were included to predict potential regulators by using iRegulon software. See experimental procedures for details. (**d**) GCN5 knockdown impaired IL1α, IL1β and GCLM expression. qPCR was performed to analyse the mRNA level in control and GCN5 knockdown cells ($n = 3$, mean ± s.e.m.). (**e**) GCN5 knockdown abolished glucose-induced H3 K9, K14, K27 and K56 acetylation. Control and GCN5 knockdown Hep3B cells were glucose-starved for 4 h and added-back glucose for 2 h before whole-cell extraction for western blot analysis. (**f,g**) NEDD4 knockdown impaired the interaction between GCN5 and H3.3. Transfected Flag-H3.3 or Flag-GCN5 in Hep3B cells was immunoprecipitated to analyse its co-immunoprecipitates by western blotting. (**h**) H3 ubiquitination deficiency impaired the interaction between GCN5 and H3.3. Hep3B cells were transfected with Flag-H3.3 WT or K23/36/37R for 36 h, then glucose starved for 4 h and added-back glucose for 2 h. Cells were then lysed for co-immunoprecipitation assay using anti-Flag antibody and subsequent western blot analysis. (**i**) GCN5 and ubiquitinated H3 form complex *in vivo*. 293T cells were transfected with Flag-GCN5 and His-Ub as indicated. Briefly, sequential purification is done by first IP with Flag antibody from whole-cell extracts in RIPA buffer. Immunoprecipitates were then released from antibody/beads by buffer A and followed by *in vivo* ubiquitination assay for endogenous H3. ChIP-qPCR data were all presented as relative enrichment to 5% input and 0.05 should be multiplied to calculate the actual enrichment percentage. All asterisks (*) represent $P < 0.05$, using Student's *T*-test.

test to evaluate the statistical relationship between H3 K9 acetylation and gene expression regulated by NEDD4. The K–S test showed a significant positive correlation ($P < 0.05$) between NEDD4-dependent H3 K9 acetylation at TSS and NEDD4-regulated gene transcription (Fig. 4c), suggesting that NEDD4 orchestrates gene transcription by regulating H3 K9 acetylation at TSS. We then used quantitative PCR (qPCR) to validate the expression of top list genes from microarray and showed that NEDD4 deficiency indeed inhibited their expression (Fig. 4d,e and Supplementary Fig. 5a–c). Among those tested genes, IL1α, IL1β and GCLM were most affected and regulated by glucose (Fig. 4d–f). ChIP-seq assay revealed that H3 K9 acetylation at TSS of IL1α, IL1β and GCLM was induced by glucose in a NEDD4-dependent manner (Fig. 4g and Supplementary Fig. 5d). Consistently, ChIP-qPCR analysis confirmed this result in both NEDD4 knockdown and $Nedd4^{-/-}$ mouse embryonic fibroblasts (MEFs) (Fig. 4h,i and Supplementary Fig. 5e). In addition, we found that RNA polymerase II binding at TSS of IL1α and IL1β was induced by glucose in a NEDD4-dependent manner (Fig. 4j). As H3.3 ubiquitination affects H3 K9/K14 acetylation, we determined whether H3.3 and its ubiquitination are required for the transcription of IL1α, IL1β and GCLM. Knockdown of H3.3 decreased the messenger RNA level of IL1α, IL1β and GCLM (Fig. 5a), and such defects could be rescued by the restoration of WT H3.3, but not H3.3 K23/36/37R mutants (Fig. 5b and Supplementary Fig. 6a). We also found that restoration of single or double mutations of H3.3 partially rescued gene expression of IL1α and IL1β (Supplementary Fig. 6b,c). Furthermore, we did not find that NEDD4 knockdown or H3.3 K23/36/37R mutation affect the incorporation of H3.3 at the TSS of IL1A, IL1B and GCLM genes (Supplementary Fig. 6e–g), excluding the possibility that those effects of H3 ubiquitination on gene transcription are due to the defect in H3.3 deposition. These data underpin the function of H3 ubiquitination by NEDD4 in transcription activation.

**H3 ubiquitination regulates H3 acetylation by recruiting Gcn5.** To decipher the underlying mechanism by which H3 ubiquitination regulates H3 acetylation, we determined whether H3 ubiquitination is crucial for the recruitment of chromatin remodelling complexes containing acetyltransferase activity. By comparing microarray data sets against published ChIP-seq tracks for various chromatin-binding factors[29], we identified GCN5, a HAT that preferentially catalyses H3 K9 and K14 acetylation in mammalian cells, may be a potential candidate to mediate the unidirectional crosstalk between H3 ubiquitination and acetylation (Fig. 5c). We then provided a series of experimental evidence to further confirm this notion. First, we found that the mRNA level of IL1α, IL1β and GCLM was reduced upon GCN5 knockdown (Fig. 5d), similar to the effect of glucose deprivation, NEDD4 knockdown and H3 ubiquitination deficiency. Second, knockdown of GCN5 selectively impaired glucose-induced H3 acetylation on K9, K14, K27 and K56 (Fig. 5e and Supplementary Fig. 7a), but failed to affect H3 acetylation on other sites and H3 methylation (Supplementary Fig. 7b), phenocopying the effect of NEDD4 knockdown or glucose starvation. Third, we found that NEDD4 knockdown impaired the interaction between H3 and GCN5 by using the reciprocal immunoprecipitation assays (Fig. 5f,g). Notably, GCN5 preferentially pulled down monoubiquitinated H3 in the co-immunoprecipitation assay and in the *in vitro* binding assay, indicating that H3 ubiquitination facilitates the recruitment of GCN5 to H3 (Fig. 5g and Supplementary Fig. 7c). Knockdown of NEDD4 also reduced the glucose-induced GCN5 recruitment to chromatin (Supplementary Fig. 7d). Fourth, we demonstrated

that glucose induced the interaction between H3 and GCN5 in a H3 ubiquitination-dependent manner (Fig. 5h). Finally, by sequential purification assay, we demonstrated that GCN5 and ubiquitinated H3 formed complex *in vivo* (Fig. 5i). To comprehensively understand the dynamic recruitment of acetyltransferase by H3 ubiquitination, we also examined the recruitment of other known H3 acetyltransferases, p300 and P300/CBP-associated factor (PCAF), by co-immunoprecipitation assay. However, we found there is no obvious difference between the interaction of p300 or PCAF with WT H3 and H3 K23/36/37R mutant (Supplementary Fig. 7e), suggesting a role of H3 ubiquitination in the HAT specificity. In addition, GCN5 knockdown did not reversely affect H3 ubiquitination (Supplementary Fig. 7f). Collectively, these results suggest that glucose-induced H3 ubiquitination by NEDD4 facilitates the complex formation between Gcn5 and H3, which is associated with glucose-induced H3 acetylation.

**H3 ubiquitination regulates tumour sphere and tumour engraftment.** To understand the biological significance of glucose-induced H3 ubiquitination by NEDD4, we applied GSEA to discover enriched gene sets in microarray data sets. We found that knockdown of NEDD4 profoundly impaired multiple cancer-related pathways using a cut-off suitable for exploratory discovery ($P < 0.05$ and $q < 0.25$, K–S test)[30] (Fig. 6a and Supplementary Fig. 8a,b), indicating the potential role of NEDD4 in cancer[13]. To evaluate the clinical relevance of this finding, we analysed the The Cancer Genome Atlas (TCGA) exon expression data sets for NEDD4 and top-ranked genes in cancer-related pathways from microarray (Supplementary Fig. 8c). We found that in multiple cancer types, their expressions were significantly ($P < 0.05$, Wilcoxon test) positively correlated (Fig. 6b and Supplementary Fig. 8d). This observation led us to hypothesize that glucose-induced H3 ubiquitination by NEDD4 may have a critical role in cancer regulation. To test this hypothesis, we determined whether the population of cells with high aldehyde dehydrogenase activity (Aldh$^+$) and the sphere-forming ability are affected by glucose-induced H3 ubiquitination by NEDD4. Notably, we found that *in vitro* tumour sphere numbers and Aldh$^+$ cell population were reduced upon glucose depletion, NEDD4 knockdown, NEDD4 tyrosine phosphorylation-deficient mutation, GCN5 knockdown, H3.3 knockdown and deficiency of H3 ubiquitination (Fig. 6c–h and Supplementary Fig. 9a–l). To further examine the *in vivo* function, 1,000 Aldh$^+$ cells were subcutaneously injected into nude mice to evaluate the *in vivo* tumour engraftment frequency. We found that knockdown of NEDD4 or H3 K23/36/37R mutation reduced the tumour incidence and average tumour size, suggesting that NEDD4-mediated H3 ubiquitination is also required for the tumorigenecity of Aldh$^+$ cells *in vivo* (Fig. 6i–n and Supplementary Fig. 10a,b). Together, these studies reveal an important role of glucose-induced H3 ubiquitination by NEDD4 in cancer development.

**Glucose/NEDD4 target genes regulate tumour sphere formation.** As H3 acetylation is associated with transcriptional activation, we speculated that the effect of this glucose-mediated signalling pathway, consisting of NEDD4, Gcn5, H3 ubiquitination and acetylation, on tumour sphere-forming cells is likely to be dependent on the H3 acetylation and transcriptional activation. We next determined which set of target genes is involved in tumour sphere formation. As we have already identified IL1α, IL1β and GCLM as important transcriptional targets of glucose-induced H3 ubiquitination (Fig. 4e,l), we investigated whether they are involved in tumour sphere formation. Although

neutralizing extracellular IL1α and IL1β individually had little impact on tumour sphere formation *in vitro*, simultaneous sequestering of IL1α and IL1β resulted in a drastic reduction of tumour sphere numbers (Fig. 7a and Supplementary Fig. 11a), suggesting that IL1α and IL1β are required, but likely to

compensate each other, for maintaining tumour sphere-formation ability. Interestingly, we also found that the treatment of IL1β could not fully rescue the defect in tumour sphere formation upon NEDD4 knockdown, despite that IL1β alone readily increased the tumour sphere numbers in control cells (Fig. 7b and

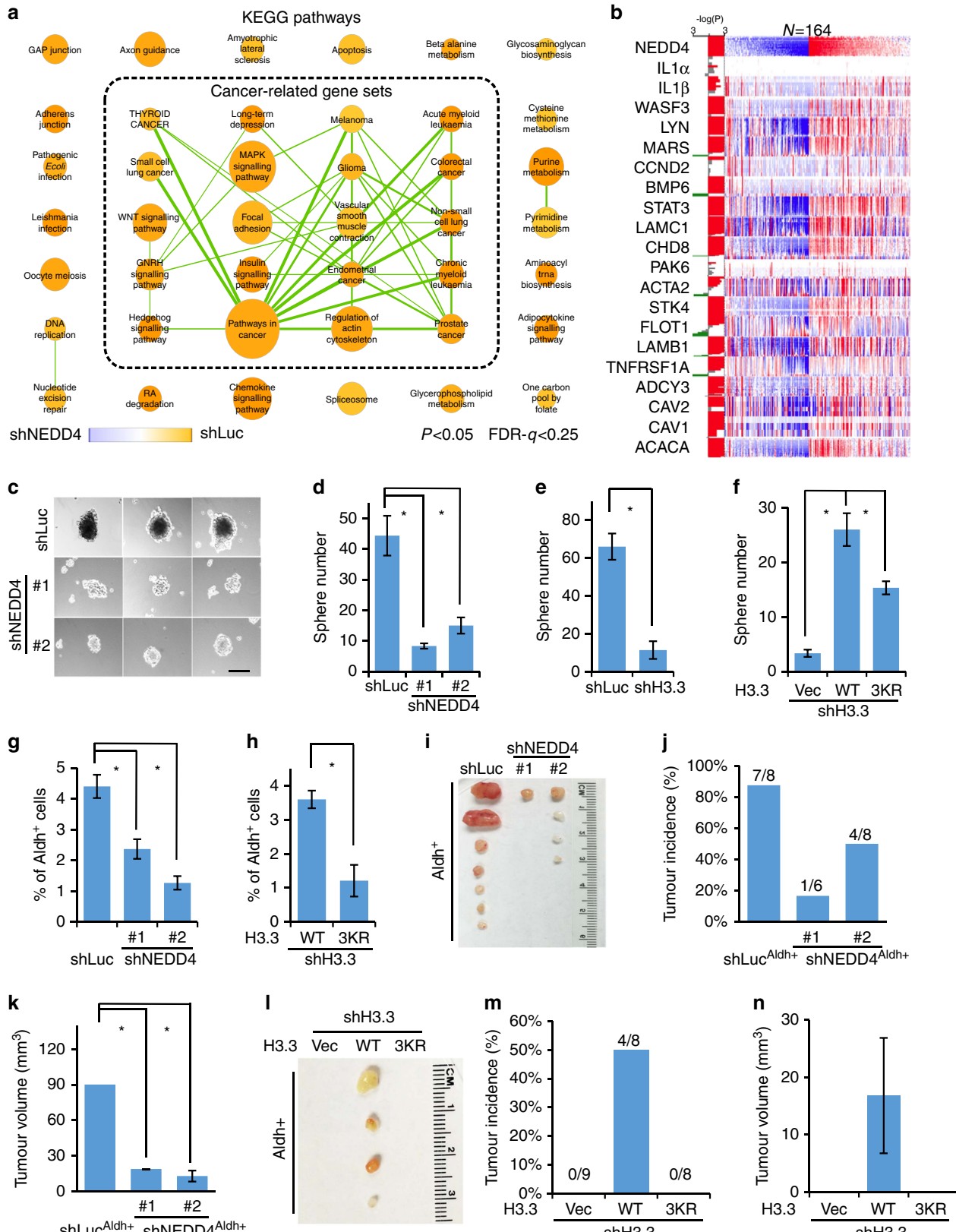

Supplementary Fig. 11b), indicating that there are other important factors involved. As excessive reaction oxygen species (ROS) has deleterious impact on cancer cells[31], we rationalized that loss of GCLM, which is a key enzyme subunit in the biosynthesis of anti-oxidant glutathione (GSH), may lead to ROS elevation and subsequent inhibition of tumour sphere formation. In line with this assumption, we found that the knockdown of NEDD4 or deficiency of H3 ubiquitination, which impaired GCLM expression, increased cellular ROS level and decreased GSH level (Fig. 7c–e). Treatment of N-acetyl-L-cysteine (NAC), a widely used ROS scavenger, not only increased tumour sphere number in control cells but also partially rescued NEDD4 knockdown phenotype (Fig. 7b and Supplementary Fig. 11b). Remarkably, co-administration of NAC and IL1β completely rescued the defect in tumour sphere formation upon NEDD4 knockdown (Fig. 7b and Supplementary Fig. 11b), suggesting both IL1β and anti-oxidant are critical for tumour sphere formation. Together, these results reveal that IL1α, IL1β and GCLM are critical downstream mediators of H3 ubiquitination signalling in cancer regulation.

## Discussion

NEDD4 in many occasions is identified as a cytosolic E3 ligase, which targets its protein substrates mostly for trafficking and proteasome degradation[13]. Our study reveals the non-proteolytic role of NEDD4 in the nucleus. We show that NEDD4 is presented in both the cytoplasm and nucleus under normal condition. Glucose treatment on glucose-starved cells elevated cellular calcium level, which is associated with NEDD4 E3 ligase activation, providing an explanation why H3 ubiquitination is enhanced by glucose treatment. Moreover, we also identify that NEDD4 monoubiquitinates H3 at multiple sites, consistent with the capability of NEDD4 in triggering monoubiquitination on its substrates as IGF1R and IRS2 (refs 15,32). Although NEDD4 has been controversially implicated in regulating AKT signalling[33,34] and AKT signalling may also participate in the regulation of H3 acetylation[35], we did not find that AKT signalling is affected by NEDD4 knockdown in our glucose treatment experiments (treating glucose in glucose-starved cells) and sphere-forming experiments (Supplementary Fig. 12a,b). Of note, AKT phosphorylation is reduced upon glucose treatment (Supplementary Fig. 12a), which is opposite to the glucose effect on H3 ubiquitination and acetylation. This result may be counterintuitive to the study showing that AKT activity is required for the H3 acetylation increase by switching from low glucose (1 mM) to high glucose (10 mM) condition[35], revealing that AKT phosphorylation and H3 acetylation in cancer may not be linked under certain circumstances. Of note, our results show

that NEDD4 knockdown does not affect acetyl-CoA level (Supplementary Fig. 3e,f), which also differs from the effect of AKT on H3 acetylation[35]. Thus, it is unlikely to be that our finding showing NEDD4 regulates H3 acetylation and tumour sphere is through AKT signalling pathway. These results further extend the versatile role of NEDD4 in cellular signalling independent of AKT.

In this study, we found that glucose could induce the tyrosine phosphorylation of NEDD4, which activates NEDD4 E3 ligase activity. However, it is unclear which upstream signalling pathway is involved in NEDD4 phosphorylation. Earlier study showed that fibroblast growth factor or epidermal growth factor could induce NEDD4 tyrosine phosphorylation through Src kinase[16]. However, Src kinase is not activated upon glucose treatment (Supplementary Fig. 12c). Previous study also showed that tyrosine kinase Yes, the closest member of Src among the Src family kinases (SFKs), is activated by glucose[36]. Consistently, add-back glucose induced Yes phosphorylation, but not Src and Fyn (Supplementary Fig. 12c). Treatment of PP2, the inhibitor for SFKs including Yes, could also inhibit the glucose-induced tyrosine phosphorylation of NEDD4 in the glucose add-back experiment (Supplementary Fig. 12d). Importantly, glucose-induced H3 ubiquitination and H3 acetylation were also inhibited by PP2 (Supplementary Fig. 12e), suggesting that SFK activation by glucose is important for glucose-induced NEDD4 activation. In the future, it is our goal to study whether Yes is a direct kinase for NEDD4 and how it is involved in the cellular responses to the changes in glucose level.

Many HATs, such as Gcn5, PCAF, p300/CBP and RTT109, are shown to catalyse H3 acetylation on the N-terminal tail with distinct preferred lysine residues[3,4]. Although histone acetylation by HATs generally promotes transcription activation, it is unclear how those HATs are selectively utilized to acetylate H3 at specific lysine residues under certain physiological cues. In this study, we found that in glucose stimulation condition, Gcn5 is important for the acetylation of K9, K14 and K27 on H3. It should be noted that this does not exclude the important role of other HATs in other physiological conditions. More importantly, H3 ubiquitination by NEDD4 may facilitate the binding of H3 with Gcn5, but not PCAF and p300, for H3 acetylation upon glucose stimulation. H3 acetylation or methylation on K23 and K36 residues, which are H3 ubiquitination sites, are not affected by glucose or NEDD4 knockdown, indicating that these lysine sites are likely to be reserved for H3 ubiquitination under glucose stimulation condition to convey specificity for HAT recruitment and acetylation/methylation on such sites are not involved in glucose-mediated signalling events. More importantly, these data suggest that the effect of H3 K23/36/37R mutation on glucose-mediated HAT recruitment and transcription are more likely to

**Figure 6 | H3 ubiquitination is required tumour sphere forming and tumour engraftment.** (**a**) Cancer-related gene sets were enriched in control versus NEDD4 knockdown Hep3B cells. KEGG pathway gene sets enriched ($P < 0.05$, $q < 0.25$ for exploratory type of study) in control or NEDD4 knockdown were presented as orange or blue circles, respectively. Gene sets with overlapping genes were connected by green lines. The weight of the circle and line are proportional to the number of genes in gene set and overlapping genes between gene sets, respectively. See experimental procedures for details. (**b**) NEDD4 expression is correlated with NEDD4 target genes in TCGA hepatocarcinoma exon expression data sets. See experimental procedures for details. (**c–f**) NEDD4, H3.3 and H3 ubiquitination are required for in vitro tumour sphere formation. NEDD4 knockdown, H3.3 knockdown or H3.3 WT or K23/36/37R restored Hep3B cells were analysed by in vitro tumour sphere-forming assay (see experimental procedures for details). Scale bar, 300 μm in length (**c**). Data were presented as the mean number of three biological replicates ± s.e.m. (**g,h**) NEDD4 and H3 ubiquitination are required for maintaining Aldh+ cell population. Control and NEDD4 knockdown or H3 WT or K23/36/37R restored Hep3B cells were stained for Aldh enzymatic activity and analysed by flow cytometry. Data were presented as the mean percentage of three biological replicates ± s.e.m. See experimental procedures for details. (**i–k**) NEDD4 knockdown reduced in vivo tumour engraftment frequency of Du145 cells. Shown were tumour image, tumour incidence and tumour size, which was presented as the mean volume of tumours (($L \times W \times W$)/2) ± s.e.m. $n = 9$ for each group and dead mice free of tumours are excluded. (**l–n**) K23/36/37R mutation reduced in vivo tumour engraftment frequency of Du145 cells. Shown were tumour image, tumour incidence and tumour size, which was presented as the mean volume of tumours (($L \times W \times W$)/2) ± s.e.m. See experimental procedures for details. All asterisks (*) represent $P < 0.05$, using Student's T-test.

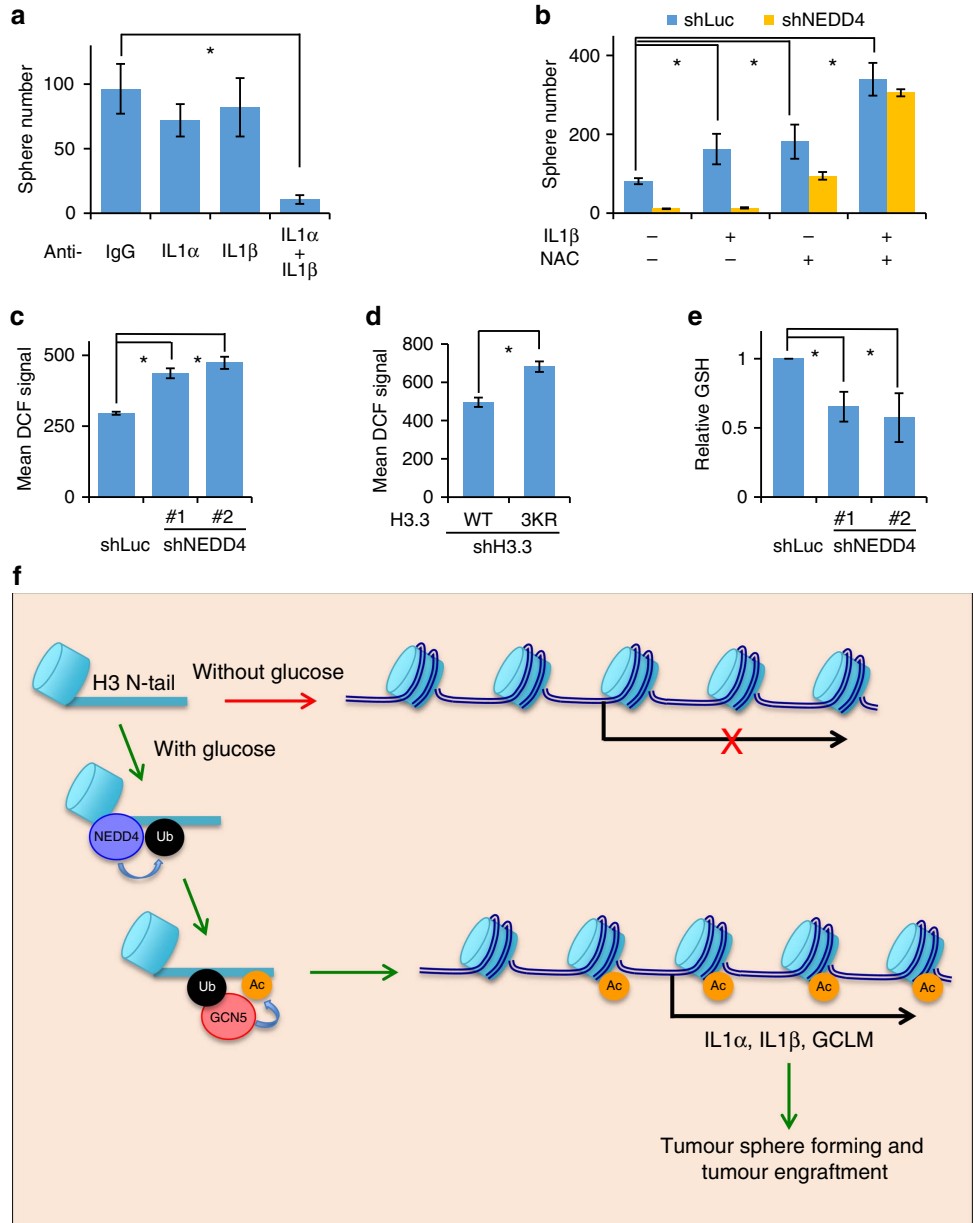

**Figure 7 | IL1α/IL1β and ROS homeostasis are critical for tumour sphere-forming ability.** (**a**) Simultaneous neutralization of IL1α and IL1β abolished *in vitro* tumour sphere formation. Anti-IL1α (1:200) and anti-IL1β (1:200) neutralizing antibodies were added to medium on day 1 and day 4 after seeding Hep3B cells for *in vitro* tumour sphere-forming assay. Data were presented as the mean number of three biological replicates ± s.e.m. See Supplementary Fig. 7a and experimental procedures for details. (**b**) IL1β and NAC co-treatment rescued the defect of NEDD4 knockdown in the *in vitro* tumour sphere formation. Control and NEDD4 knockdown Hep3B cells were treated with recombinant IL1β and/or NAC (0.5 mM) in the *in vitro* tumour sphere-forming assay. Data were presented as the mean number of three biological replicates ± s.e.m. See Supplementary Fig. 7b and experimental procedures for details. (**c,d**) NEDD4 and H3 ubiquitination are required for the maintenance of cellular ROS. Control and NEDD4 knockdown or H3.3 WT or K23/36/37R restored Hep3B cells were stained by DCFDA for cellular ROS and subjected to flow cytometry analysis. Data were presented as the mean DCFDA signals of three biological replicates ± s.e.m. See experimental procedures for details. (**e**) GSH level is reduced in NEDD4 knockdown cells. Control and NEDD4 knockdown Hep3B cells were collected and GSH levels were measured by a colorimetric enzymatic reaction. Data were presented as the mean value of three biological replicates ± s.e.m. See experimental procedures for details. (**f**) Model of glucose-induced H3 mono-ubiquitination by NEDD4 and subsequent GCN5-mediated H3 acetylation, which regulates tumour sphere forming and tumour engraftment through transcription activation of genes, such as *IL1A*, *IL1B* and *GCLM*. N-tail represents histone H3 N-terminal tail. All asterisks (*) represent P < 0.05, using Student's *T*-test.

be through reducing the H3 ubiquitination, but not acetylation and methylation on the same sites. It is also yet to be studied how H3 ubiquitination facilitates the recruitment of Gcn5 in the future. There are several possibilities to explain how H3 ubiquitination regulates the recruitment of Gcn5: (i) ubiquitinated H3 may serve as a substrate with higher affinity to the enzymatic domain of Gcn5 as compared with non-

ubiquitinated H3; (ii) Gcn5 may recognize ubiquitin conjugated H3 through previously undefined sequences that could interact with ubiquitin; (iii) as Gcn5 often forms complex with chromatin-associated complexes, other subunits in those complexes may contain ubiquitin binding motifs and may recognize ubiquitin conjugated H3 upon glucose treatment and facilitate the recruitment of Gcn5. This also leads to another

question that in the glucose context, whether Gcn5 alone or chromatin remoulding complex associated Gcn5 mediated the H3 acetylation and this could be tested in the future that whether H3 ubiquitination affects the recruitment of Gcn5-associated chromatin remoulding complexes. Together, our findings collectively provide more insights into the regulation of histone acetylation by Gcn5 specifically under glucose-induced condition.

In vitro tumour sphere can be formed by a subgroup of cancer cells that possesses normal stem cell-like characteristic and likely to be responsible for tumour initiation, drug resistance, relapse and metastasis[37–41]. These traits are generally originated from abnormal genetic or epigenetic alteration, including histone modification. Our study provides evidence that glucose-mediated signalling, consisting of NEDD4-mediated H3 ubiquitination and subsequent Gcn5-mediated H3 acetylation, plays important roles in maintaining such population of cancer cells. Notably, we experimentally demonstrate that maintenance of tumour sphere-forming cells are dependent on both IL1α/IL1β and cellular ROS homeostasis conveyed by GCLM, which are all glucose-inducible genes. Thus, in the glucose context, our results indicate that increased glucose demand in cancer cells is a prerequisite not only for producing more anti-oxidants, energy and cellular building blocks, but also for activating important cellular signalling to maintain H3 ubiquitination and acetylation, thereby driving transcriptional activation of crucial tumour sphere factors. Our findings further underscore the important role of glucose in tumour sphere-forming cells and provide a revolutionized view for glucose in the maintenance of tumour sphere-forming cells through activating and maintaining the expression of IL1α/IL1β and GCLM. Thus, targeting glucose/NEDD4-dependent H3 ubiquitination and subsequent transcription of IL1α/IL1β and GCLM may be an effective way to target cancers.

In summary, in addition to acetyl-CoA, our study identifies that NEDD4-dependent H3 ubiquitination is important for the rapid induction of H3 acetylation by glucose, which plays a critical role in transcriptional activation of tumour sphere factors for the maintenance of tumour sphere formation (Fig. 7f). Hence, targeting this newly identified signaling pathway, such as NEDD4, Gcn5 or their transcriptional targets including IL1α, IL1β and GCLM may be promising approaches for cancer therapy.

## Methods

**Cell culture and viral transduction.** Hep3b, HEK293T and Du145 cell lines were purchased from ATCC and cultured in DME/F12 medium (GE Healthcare HyClone) in 10% fetal bovine serum (Sigma Aldrich), Penicillin Streptomycin and L-glutamine. WT and $Nedd4^{-/-}$ mouse embryonic fibroblasts cultured in DMEM medium containing 10% fetal bovine serum were obtained from Dr Xinjiang Wang. For the add-back of glucose procedure, 80% confluent cells were cultured in DMEM medium without glucose (Invitrogen) for 4 h and resuppled with glucose $(4.5 \, g \, l^{-1})$ for indicated times. 293T cells were transfected by standard calcium phosphate method. Other cell lines were transfected using Lipofectamine 3000 (Invitrogen) as per the manufacturer's protocol and, if indicated, cultured in medium containing Hygromycin B $(100 \, \mu g \, ml^{-1})$ to establish stable cell lines. For lentiviral infection, 293T cells were co-transfected with lentiviral plasmid (pLKO-puro), packing plasmid (deltaVPR8.9) and envelope plasmid (VSV-G). Two days after transfection, medium containing virus particles were used to infect mammalian cell lines. All the infected cells were cultured in the medium containing $2 \, \mu g \, ml^{-1}$ puromycin for 1 week before further analysis. Short interfering RNA for NEDD4, H3.3 and GCN5 were purchased from Sigma Aldrich.

**Reagents.** Flag-H3.3 plasmid and lentiviral Flag-H3.3 plasmid were obtained by inserting human H3.3 open reading frame into pCDNA3.1-hygro or pLKO-as3w-puro vectors, respectively. All H3.3 mutation constructs (K4R, K9R, K14R, K18R, K23R, K27R, K36R, K37R, K36/37R, K23/36/37R, K56R, K64R, K79R, K115R and K122R) were generated by site-directed mutagenic PCR according to the kit manual (Stratagene). His-ubiquitin plasmid was previously described[42]. HA-BMI-1 and HA-RNF2 were obtained from Dr Shiaw-Yih Lin. HA-NEDD4 and HA-UHRF-1 plasmids were from Drs Pier Paolo Pandolfi and Hung-Ying Kao, respectively. HA-CBL-A, HA-CBL-B and HA-CBL-C were received from

Dr Stanely Lipkowitz. Flag-HA-RNF8 was kindly provided by Dr Junjie Chen. V5-NEDD4 WT, Y43/585F and Y43/585E mutant are from Dr Daniela Rotin. Flag-NEDD4L, NEDL1 and NEDL2 are from Dr Wesley Sundquist. Flag-WWP1 and WWP2 are from Dr Wenyi Wei. The following antibodies were used in this study: anti-H3 (Abcam, ab12079, 1:5,000), anti-H3 pan-ac (Active Motif, 39139, 1:2,000), anti-H3 K4ac (Active Motif, 39381, 1:2,000), anti-H3 K9ac (Active Motif, 39917, 1:3,000), anti-H3 K14ac (Active Motif, 39697, 1:2,000), anti-H3 K18ac (Active Motif, 39755, 1:2,000), anti-H3 K23ac (Active Motif, 39131, 1:2,000), anti-H3 K27ac (Active Motif, 39133, 1:2,000), anti-H3 K36ac (Active Motif, 39379, 1:2,000), anti-H3 K56ac (Active Motif, 61061, 1:2,000), anti-H3 K4me3 (39915, 1:5,000), anti-H3 K9me3 (Active Motif, 39765, 1:5,000), anti-H3 K27me3 (Active Motif, 39155, 1:5,000), anti-H3 K4me2, K9me2, K27me2, K36me2, K79me2 (Cell Signaling Technology, 9847, 1:5,000), anti-H3s10p (Abcam, ab5176, 1:1,000), anti-H3.3 (EMD Millipore, 09-838, 1:1,000), anti-NEDD4 (Novus, NBP1-40112, 1:4,000), anti-GCN5 (Active Motif, 39975, 1:1,000), anti-USP22 (Abcam, ab4812, 1:2,000), anti-Flag (Sigma, 1:2,000), anti-HA (Covance, 1:2,000), anti-Actin (Sigma, 1:10,000), anti-IL1α (Abcam, ab17281), anti-IL1β (Abcam, ab2105) and anti-IgG heavy chain HRP (Sigma Aldrich, a1949, 1:1,000). Uncropped scans for western blottings presented in the main figures are provided in Supplementary Fig. 13. The following recombinant proteins were used in this study: active NEDD4 (Millipore), histone octamer (EMD Millipore), E1 enzyme (Sigma Aldrich), UbcH4 (EMD Millipore), UbcH5a (EMD Millipore), UbcH5b (EMD Millipore), UbcH5c (Boston Biochem), UbcH6 (EMD Millipore), UbcH7 (Boston Biochem), ubiquitin (Boston Biochem), GST-ubiquitin (Millipore), GCN5 (Novus) and IL1β human (Sigma Aldrich). The following chemicals were used in this study: NAC (Sigma Alrich), N-ethylmaleimide (NEM) (Calbiochem), DCFDA (Invitrogen) and PP2 (Sigma).

**In vivo ubiquitination assay.** Briefly, 293T cells were transfected with his-Ubiquitin and other indicated plasmids for 36 h and harvested by denaturing buffer A (6 M guanidine-HCl, 0.1 M Na$_2$HPO$_4$/NaH$_2$PO$_4$, 10 mM imidazole pH 8). Pre-washed Ni-NTA Agarose beads (QIAGEN) were incubated with cell lysates for 3.5 h to pull-down his-Ubiquitin and his-Ubiquitin-conjugated proteins. Beads were then washed by buffer A and buffer T1 (25 mM Tris-Cl, 20 mM Imidazole pH 6.8) and analysed by western blotting.

**In vitro ubiquitination assay.** Three micrograms of recombinant histone octamer and 5 μg active form of NEDD4 were incubated with 0.5 μg E1 activating enzyme, 1.5 μg Ubiquitin, 0.5 μg various E2 enzymes UBCH4, UBCH5a, UBCH5b, UBCH5c, UBCH6 or UBCH7 and 2.5 mM ATP in reaction buffer (1.5 mM MgCl$_2$, 5 mM KCl, 1 mM dithiothreitol (DTT), 20 mM HEPES pH7.4) in a total 20 μl reaction volume at 37 °C for 3 h. Ubiquitination on substrate was then detected by western blot analysis. Flag-H3 WT and K23/36/37 containing nucleosomes were purified by nucleosome preparation kit (active motif) followed with Flag-tag affinity purification and dialysed/concentrated (Amicon Ultra) against in vitro ubiquitination buffer.

**Whole-cell extracts for histone modification detection.** Whole-cell extracts were prepared by boiling the cell pellets in SDS sample buffers for 10 min.

**Cellular fractionation and chromatin fractionation.** Cytosol and nucleus were purified using standard protocol. Briefly, cells were re-suspended in hypertonic buffer (10 mM Tris pH 7.6, 10 mM MgCl$_2$, 0.1% NP-40), then lysed using Dounce homogenizer and centrifuged at 1,000 g. Pellets containing nucleus were washed twice with hypertonic buffer. Supernatant containing cytosol was further cleared by centrifuging at 12,000 g. Chromatin fractionation was performed as described[43]. Briefly, cells were first lysed with buffer A (50 mM Hepes pH 7.9, 10 mM KCl, 1.5 mM MgCl$_2$, 0.34 M Sucrose, 10% Glycerol, 1 mM DTT, 0.1% Triton X-100, NEM, protease inhibitor cocktail (Biotool) and phosphatase inhibitor cocktail (Biotool)) on ice for 30 min. After centrifugation at 1,000 g, pellets including the nucleus were further lysed with buffer B (3 mM EDTA, 0.2 mM EGTA, 1 mM DTT, NEM, protease inhibitor cocktail (Biotool) and phosphatase inhibitor cocktail (Biotool)). After centrifugation, pellets containing the chromatin were washed and sonicated in SDS sample buffer for western blot analysis.

**Immunoprecipitation.** To immunoprecipitate proteins, cells were lysed and sonicated in RIPA buffer (50 mM Tris pH 7.4, 150 mM NaCl, 0.1% SDS, 0.5% sodium deoxycholate, 1% NP-40, 1 mM EDTA, protease inhibitor cocktail (Biotool), phosphatase inhibitor cocktail (Biotool) and NEM). Cell lysates were incubated at 4 °C with antibody overnight and agarose protein A/G beads for 3 h, and then beads were washed with RIPA buffer for five times before eluting with SDS sample buffer.

**Assay for endogenous ubiquitinated proteins.** Briefly, cells were collected and boiled in SDS lysis buffer (2% SDS, 150 mM NaCl, 50 mM Tris-HCl, pH 7.4, protease inhibitor cocktail) for 10 min. Lysate was sonicated and diluted 10 times with dilution buffer (50 mM Tris-HCl pH 7.4, 150 mM NaCl, 1 mM EDTA, 0.5% sodium deoxycholate, 1% NP-40, protease inhibitor cocktail and NEM). Samples

were then incubated on ice for 30 min and cleared by centrifuging. Immunoprecipitation assay using anti-Ub (FK2) antibody was performed as described above and H3 ubiquitination was detected by western blotting using anti-H3 antibody.

**ChIP followed by qPCR.** ChIP assay was performed as described[44] with some modifications. Cells were cross-linked in culture media (with 1% formaldehyde) with gentle shaking for 10 min at room temperature and stopped by adding glycine to a final concentration of 0.125 M. Cells were washed with PBS three times and nuclei were isolated. Nuclei were then lysed in RIPA buffer with proteinase inhibitor cocktails and sonicated using Bioruptor to shear genomic DNA, to a range of 200–1,000 bps. Lysates were cleared and blocked with BSA (final concentration of 1 mg ml$^{-1}$) and salmon sperm DNA (final concentration of 0.3 mg ml$^{-1}$). Pre-cleaned lysates were immunoprecipitated with various antibodies, followed by adding protein A/G beads. Beads were washed and eluted by 400 µl elution buffer (with 0.1 M NaHCO$_3$, 1% SDS). The DNA was directly recovered by gel extraction kit (Omega) and analysed by real-time qPCR.

**ChIP followed by sequencing.** ChIP assay was performed using ChIP-IT High Sensitivity kit (Active Motif) according to the manufacturer's manual. The Illumina compatible libraries were prepared using DNA Library preparation kit (KAPA, KK8232), as per the manufacturer's protocol. In brief, DNA was fragmented to a median size of 150 bp by sonication. Fragmented DNA ends were polished and 5′-phosphorylated. After addition of 3′-A to the ends, indexed Y-adapters were ligated and the samples were PCR amplified. The resulting DNA libraries were quantified and validated by qPCR, and sequenced on Illumina's HiSeq 2000 in a single-read format for 36 cycles. The resulting BCL files containing the sequence data were converted into '.fastq.gz' files and individual sample libraries were demultiplexed using CASAVA 1.8.2 with no mismatches.

**ChIP-seq data analysis.** Briefly, 36 nucleotides sequencing data (.fastq.gz) were unachieved and imported to local galaxy project instance[45–47]. Sequences for each sample were concatenated and 3 nucleotides from both 5′- and 3′-end were trimmed off. The processed data were aligned to the hg19 (human) assembly using Bowtie2 (ref. 48). For each sample, the ChIP-seq peak profiles were obtained by normalizing ChIP data to input data (mappable reads were normalized to 1 × genome coverage of hg19) using BamCompare tool in DeepTools[49]. The data were visualized by preparing custom track hubs on the UCSC genome browser. Global average profiles at TSS or enhancer regions were calculated by ComputeMatrix tool in DeepTools[49] and visualized by Microsoft Excel. Known enhancer regions were defined according to Broad ChromHMM tracks for HepG2 cells. Differential ChIP-seq peaks between samples were identified by Diffreps using G-test[50]. Genes, the TSS of which was located within differential peaks, were then listed for subsequent Venn diagram visualization.

**RNA extraction and cDNA synthesis.** Cells were lysed in TRIZOL reagent (Invitrogen) and extracted by chloroform. Total RNA were then precipitated by isopropanol and washed with 70% ethanol. Complementary DNAs were synthesized according to the standard M-MLV reverse-transcriptase protocol. Briefly, total RNA were mixed with oligo(dT), dNTP, M-MLV reverse transcriptase (Invitrogen) in the M-MLV buffer and the reaction was performed using a thermo cycler.

**Microarray and data analysis.** Total RNA was extracted and purified using RNeasy mini kit (QIAGEN) according to the manufacturer's manual. Microarray analysis was performed for total RNA on Illumina HumanHT12v4 platform following Illumina's standard procedure. All data sets were normalized based on the mean value and differentially expressed genes were ranked by T-test. GSEA analysis was performed using GSEA software with Kyoto Encyclopedia of Genes and Genomes (KEGG) pathway gene sets and gene ontology (GO) term pathway gene sets, respectively. Significantly enriched gene sets ($P < 0.05$, $q < 0.25$) were visualized using Cytoscape[30]. TCGA exon expression data sets (Illumina) for various cancer types were visualized in UCSC cancer genome browser. Cases with highest and lowest 30% expression of NEDD4 were included to evaluate the correlation between the expression of NEDD4 and other genes by Wilcoxon test (Bonferroni correction). Exons show significant ($P < 0.05$) correlations with NEDD4 expression were shown either in red (positive correlation) or green (negative correlation) in the histogram.

**In vitro binding and native gel analysis.** Recombinant full-length GCN5 and ubiquitin were mixed in the in vitro binding buffer (25 mM HEPES pH 7.4, 125 mM KCl, 5 mM MgCl$_2$, 0.5 mM EDTA, 0.5% Triton X-100) and incubated on ice for 30 min. Samples were then mixed with sample buffer (62.5 mM Tris-HCl pH 6.8, 25% glycerol, 1% Bromophenol blue) or SDS sample buffer and then separated by native PAGE or SDS–PAGE, respectively. GCN5 and ubiquitin proteins were then detected by western blotting. Flag-Gcn5-containing complex was pulled down from 293T cells expressing Flag-Gcn5 by Flag-tag, eluted by Flag-peptide and dialysed/concentrated (Amicon Ultra) against in vitro binding buffer. In vitro

ubiquitinated Flag-H3.3 was also pulled down by Flag-tag, eluted by Flag-peptide and dialysed against in vitro binding buffer.

**Tumour sphere formation and tumour engraftment assay.** For in vitro tumour sphere-forming assay, 5000 cells were seeded in the ultra-low attachment 6-well plate (Corning Life Science) and cultured in tumour sphere-forming medium (DME/F12 supplemented with 5 µg ml$^{-1}$ Insulin, 0.05 µg ml$^{-1}$ human epidermal growth factor (hEGF) and 0.5 µg ml$^{-1}$ Hydrocortisone). Cells were incubated at 37 °C for 13 days and spheres larger than 100 µm were counted. For in vivo tumour engraftment assay, cancer cells were first stained using ALDEFLUOR Kit (StemCell Technology) following the manufacturer's manual. Cells with top 2.5% Aldh enzymatic activity were then isolated by cell sorting using flow cytometry. One thousand isolated cells were subcutaneously injected into each nude mouse (NCRNU-F/Homozygous, Taconic Farms, 5-week-old, total 54 mice were used) and monitored for tumour growth for 70 days. Mice died tumour free were excluded from the final results. No statistical method was used to predetermine the sample size. No specific randomization method was used. The person who performed the subcutaneous injection is blinded to the group allocation of cancer cell line. All animal experiments were performed under Institutional Animal Care and Use Committee approval protocol.

**ROS and GSH detection.** To detect cellular ROS, cells were collected and incubated with DCFDA at 37 °C for 30 min and then subjected to flow cytometry analysis. ROS levels were calculated as the mean fluorescence signal. Cellular GSH level was determined by Glutathione assay kit (Sigma Aldrich, CS0260).

**Data availability.** All ChIP-seq and microarray data sets were deposited to Gene Expression Omnibus in the accession number of GSE66340, GSE90119 and GSE66341. All other remaining data are available within the article and its Supplementary Information files or available from the authors upon request.

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

## Acknowledgements
We thank Drs Pier Paolo Pandolfi, Mien-Chie Hung, Wei Gu, Stanley Lipkowitz, Junjie Chen, Shiaw-Yih Lin, Xinjiang Wang, Wenyi Wei, Daniela Rotin, Wesley I. Sundquist and Hung-Ying Lin for cell lines and reagents, members from Dr Hui-Kuan Lin's laboratory for their valuable comments and suggestions, and Sequencing and Microarray core Facility (SMF) of MD Anderson Cancer Center for sequencing services. This work was supported by start-up funds from Wake Forest School of Medicine, the MD Anderson Prostate SPORE Development Award, NIH grants and CPRIT grant. SMF is supported by Core Grant CA016672. X.Z. is supported by Rosalie B Hite graduate fellowship in cancer research.

## Author contributions
X.Z. and H.-K.L. designed the experiments and wrote the manuscript. X.Z., B.L., A.H.R., X.X., P.-C.C., G.J., F.H., C.-Y.W., J.L., A.Z. and B.-S.P. performed the experiments. C.-Y.H., F.-J.T., C.H.T. and C.L. provided scientific insights and suggestions. All authors discussed and read the manuscript.

## Additional information

**Competing interests:** The authors declare no competing financial interests.

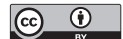

