## [Peer Review File · Nature Communications]

Reviewers' comments:

Reviewer #1 (Remarks to the Author):

In the manuscript, Zhang et al propose an interesting model that the NEDD4 and GCN5 cooperatively regulate glucose-induced histone modifications and induce gene expression involved in tumorigenesis. The authors indicate that glucose stimulation triggers NEDD4-mediated H3 ubiquitination, mostly on K23/K36/K37 residues, which recruits GCN5 to the ubiquitinated H3 to subsequently promote H3 acetylation. The histone modifications are crucial for NEDD4-dependent transcriptional regulations including the gene expression of IL1alpha, IL1beta and GCLM, among the most affected NEDD4-inducible genes contributing to sphere formation. Overall, the conclusions are supported by clear results obtained from comprehensive approaches and in-depth data analysis; therefore this study provides a new insight into the role of NEDD4 in global transcriptional regulations and tumorigenesis.

Specific comments:

In Figure 1g, the in vitro ubiquitination is not very compelling. The authors should provide more convincing data to confirm their hypothesis on the direct H3 modification by NEDD4. The assay needs to be repeated by including ubiquitination-deficient KR mutants.

What signaling pathways are involved in glucose-induced NEDD4 activation under oncogenic condition? Additionally, related to the glucose-mediated regulation of NEDD4, does glucose stimulation affect NEDD4 subcellular localization? Specifically, does NEDD4 translocate into nucleus or bind to chromatin under high glucose condition?

Reviewer #2 (Remarks to the Author):

Summary of key results:

Zhang et al report ubiquitination at histone H3 residues lysine (K) 23, 36 and 37 by E3 ligase NEDD4, and regulation of this novel histone modification by glucose. They further demonstrate that this histone H3 ubiquitination recruits Gcn5 acetyltransferase for H3K9 acetylation during transcription. Providing insight into the functions for NEDD4-mediated H3 K23/36/37 ubiquitination, the authors demonstrate a role for NEDD4 in regulating IL1 α , IL1 β and GCLM gene expression linked to tumorigenesis.

Originality and interest

While histones H2A and H2B ubiquitination are extensively studied, occurrence and functional contributions of ubiquitination at other histones remains poorly explored. Therefore, this manuscript reports a very intriguing set of findings related to novel ubiquitination modifications at histone H3. The authors have done a commendable job of characterizing the novel marks and provided functional assays linking the marks to a biological process. However, several important concerns need to be addressed prior to acceptance for publication.

Data & methodology: validity of approach, quality of data, quality of presentation

Appropriate use of statistics and treatment of uncertainties

Conclusions: robustness, validity, reliability

Suggested improvements: experiments, data for possible revision

Major Concerns

1) NEDD4 is a well-known cytosolic protein, but the authors are reporting novel nuclear functions for this protein. Therefore, additional data are needed to convincingly demonstrate that NEDD4 localizes to chromatin to perform H3 K23/36/37 ubiquitination. Moreover, is the H3 K23/36/37 ubiquitination nuclear or cytosolic? Does it occur on chromatin and co-localize with H3K9

acetylation? While authors have provided genome-wide data for glucose-regulated H3K9 acetylation, occupancy data for H3K23/36/37 ubiquitination to implicate these marks as a bona fide regulatory event during transcription is clearly absent. While the authors might argue the non-availability of antibodies recognizing H3 K23/36/37 ubiquitination for chromatin immunoprecipitation, it is certainly possible to determine the genome-wide occupancy of ubiquitinated form of a chromatin protein using target site(s) mutant and mono/poly-ubiquitin-specific antibody (FK2 - already used in this manuscript).

2) While it is commendable that the authors acknowledge the possibility that K23/36/37 ubiquitination could also occur on canonical histones H3.1 and H3.2, they do not show any test or evidence for the presence or absence of histone H3.1 or H3.2 K23/36/37 ubiquitination, even though the manuscript title implies all H3 ubiquitination (and not just H3.3 ubiquitination) playing a role in regulating H3 acetylation and tumorigenesis. It is important for the authors to test K23/36/37 ubiquitination in histones H3.1 and H3.2 and their regulation, if any, by NEDD4 and glucose. Since H3.3 is the transcription-coupled histone H3 variant, experiments testing K23/36/37 ubiquitination in histones H3.1 and H3.2 could provide the additional compelling evidence to strongly link ubiquitination of K23/36/37 to gene transcription.

Other concerns:

1) The authors state "E3 ubiquitin ligase NEDD4 ubiquitinates histone H3 on previously unstudied lysine (K) 23/36/37 residues". While it is true that histone H3K37 residue is poorly studied, H3K23 and H3K36 are well-studied histone residues in multiple eukaryotes in the context of H3K23 acetylation and H3K36 methylation and well connected to gene transcription. Therefore, saying H3 K23 and K36 residues are unstudied is very misleading.

2) While glucose add-back following starvation demonstrates its role in H3 ubiquitination, an experiment including the non-hydrolysable 2-deoxy-glucose could be included to test whether it is glucose or its breakdown that acts as an activator of H3 ubiquitination.

3) While the authors have tested H3 ubiquitination under multiple conditions, how H3 ubiquitination is regulated during various stages of cell cycle is missing. In general, upon reporting a novel histone modification, its establishment and regulation are usually tested in a cell synchronization-release experiment. Also, given that the authors are linking H3 ubiquitination to gene transcription, does inhibiting transcription (using amanitin, DRB, flavipiridol, etc.) adversely affect H3 ubiquitination levels?

4) Regulation of H3 ubiquitination by NEDD4 is convincing. Therefore, on a minor note, the human genome has >600 putative E3 ligases. The authors state 'By screening a panel of ubiquitin E3 ligases', but it is not clear what rationale was employed to select these candidate E3 ligases.

5) When the authors examine ubiquitination levels in vivo, they claim it to be an "endogenous ubiquitination assay". This is very misleading.

6) Figure 2 e-f, it is not clear if these two panels represent two independent experiments? Also, please edit to read, as 'H3 K23/36/37R is defective in H3 K9/K14 acetylation'.

7) The authors claim, "glucose-induced H3 ubiquitination by NEDD4 selectively regulates H3 acetylation" on page 10 and also in the title. However, the manuscript revolves primarily around H3K9 acetylation. In Figure 2, multiple histone acetylation (ac) marks in addition to H3K9ac are adversely affected upon knockdown of NEDD4. This data suggests that a sub-set of histone H3 acetylation (but not all H3 acetylation) is regulated by NEDD4. The authors do not provide any reasoning for why they chose to restrict their study to H3K9ac, especially since NEDD4 knockdown in fact completely abolishes H3K14ac!

8) In Figure 2, panels a - b, why is the signal for H3K36ac so variable?

9) In Figure 4d, expression of many histone genes are reduced following knockdown of NEDD4. A closer inspection of various H3 transcript levels and H3.1, H3.2 or H3.3 protein levels (using a more dynamic range) are needed to eliminate the possibility that the reduction or loss of ubiquitination is a consequence of reduced transcript/protein levels.

10) In Figure 5i, the authors show IP/pull-down assays to demonstrate the interaction between GCN5 and ubiquitinated H3. A reciprocal complementary test of H3ubi interaction with Flag-Gcn5 could also be included. As a direct test of their recruitment model, the authors could test the binding of GCN5 and/or GCN5-containing complex interaction to NEDD4-ubiquitinated nucleosomes

in vitro, and include nucleosomes with H3 K23/36/37R as a control to further demonstrate the specificity for the interaction.

Appropriate use of statistics and treatment of uncertainties

No concerns

References: appropriate credit to previous work?

Yes

Clarity and context: lucidity of abstract/summary, appropriateness of abstract, introduction and conclusions

Please refer to the comments above to improve the clarity and provide more data to arrive to support the conclusions of the manuscript.

Reviewer #3 (Remarks to the Author):

Zhang et al present a study investigating epigenetic regulation in cancer through NEDD4-dependent H3 ubiquitination. The authors find that E3 ubiquitin ligase NEDD4 is induced by glucose to ubiquitinate histone H3 on previously unstudied lysine sites K23, K36, and K37. Using ChIP-seq, the authors discover that NEDD4 regulates glucose-induced H3K9 acetylation at the transcription starting site and enhancer regions. Further, authors claim that NEDD-mediated H3 ubiquitination regulate tumor sphere formation by inducing transcription of target genes IL1 α , IL1 β , and GCLM.

Major comments:

1. In Figure 2a, the authors investigate glucose-induced H3 acetylation at H3 ubiquitination sites identified in large-scale quantitative proteomics studies shown in Figure 1a. However, not all sites are included: K37, K79 and K122 are missing. Why were these sites not investigated?
2. Statistical analysis is missing for Figures 4e, 4f, 4h, 4i, 4j, 5a, 5b, 5d, 7c, 7d, and 7e. Therefore, it is not clear whether the observed differences are statistically significant.
3. To confirm the potential role of NEDD4 in cancer, the authors use Hep3B cells for in vitro tumor sphere formation. Why were Hep3B cells used? At least two more human cancer cell lines should be used to convincingly demonstrate that NEDD4 plays a role in tumorigenesis.
4. The authors do not specify which cancer cells were used for the in vivo tumor engraftment assay. How were these cells selected?

Minor comments:

1. Aldh should be defined in the manuscript.
2. There are several typos throughout the manuscript. Some examples include:
 - a. "... (HATs) that are often resides..."
 - b. "...which is also differ from..."
 - c. "S.E.M.." Only one period should be used at the end of each sentence.

Reviewer #4 (Remarks to the Author):

The manuscript by Zhang et al. entitled "H3 ubiquitination by NEDD4 regulates H3 acetylation and tumorigenesis" describes the regulation of H3 acetylation by ubiquitination during glucose stimulation. The authors demonstrates glucose-induced ubiquitination of H3 by NEDD4 and involvement of tyrosines 43 and 585 phosphorylation during the process (Figure 1). They further

show that the NEDD4-dependent ubiquitination of the histone H3 variant H3.3 at lysines 23, 36 and 37 is critically required for the acetylation of H3K9, K14, K27 and K56 (Figure 2). Knocking down of NEDD4 compromises glucose-induced H3 acetylation at TSS and enhancer regions (Figure 3). Histone H3 acetylation induced by glucose at a large fraction of promoters and enhancers is dependent on the presence of activity of NEDD4 (Figure 4). The authors show that the NEDD4-mediated ubiquitination of H3 recruits GCN5, which is responsible for the glucose-induced H3 acetylation (Figure 5). They further demonstrate the involvement of the NEDD4 pathway in controlling tumor cell growth both in vitro and in a mouse model (Figure 6).

Comments: Histones are known to be heavily modified by acetylation, methylation and ubiquitination etc. Cross-talks between different modifications are important for regulation of transcription. In this study, the authors characterized several H3 ubiquitinations previously revealed by proteomics studies. They identified the enzyme NEDD4 that is responsible for the ubiquitination and its function in facilitating H3 acetylation during glucose stimulation and further demonstrate its in vivo function controlling tumor cell growth. This is an interesting story, which provides novel information. The experiments are well designed and the data are clean. The manuscript is well written. I feel the manuscript should be accepted for publication with a few minor modifications.

1. Figure 5c: the procedure discovering the potential regulators is not clearly described.
2. Figure S3I: Is this experiment done once only? No error bars are included.
3. Page 10. Line 7 from top: the reference for H3K9 acetylation at TSS is not the most appropriate one. The following papers should be cited: (Genes Dev. 2005 Mar 1;19(5):542-52.; Nat Genet. 2008 Jul;40(7):897-903.)

Responses to reviewer's comments

We thank for all reviewers' positive and constructive feedbacks on our manuscript. We have now experimentally addressed all the specific concerns raised by three reviewers. Please see below for responses to each reviewer's specific comments.

Reviewers' comments:

Reviewer #1 (Remarks to the Author):

In the manuscript, Zhang et al propose an interesting model that the NEDD4 and GCN5 cooperatively regulate glucose-induced histone modifications and induce gene expression involved in tumorigenesis. The authors indicate that glucose stimulation triggers NEDD4-mediated H3 ubiquitination, mostly on K23/K36/K37 residues, which recruits GCN5 to the ubiquitinated H3 to subsequently promote H3 acetylation. The histone modifications are crucial for NEDD4-dependent transcriptional regulations including the gene expression of IL1alpha, IL1beta and GCLM, among the most affected NEDD4-inducible genes contributing to sphere formation. Overall, the conclusions are supported by clear results obtained from comprehensive approaches and in-depth data analysis; therefore this study provides a new insight into the role of NEDD4 in global transcriptional regulations and tumorigenesis.

We thank for the reviewer #1 to recognize our proposed model as insightful, interesting and well-supported by our experimental data.

Specific comments:

In Figure 1g, the in vitro ubiquitination is not very compelling. The authors should provide more convincing data to confirm their

hypothesis on the direct H3 modification by NEDD4. The assay needs to be repeated by including ubiquitination-deficient KR mutants.

As suggested, we have optimized the experimental conditions and shown more convincing in vitro ubiquitination data in fig. 1g. Following the logic flow of our story, the in vitro ubiquitination assay containing the KR mutants was placed after the discovery of H3 K23/36/37 sites in fig. s3m. The data reveal that the KR mutant is deficient for in vitro ubiquitination by NEDD4.

What signaling pathways are involved in glucose-induced NEDD4 activation under oncogenic condition? Additionally, related to the glucose-mediated regulation of NEDD4, does glucose stimulation affect NEDD4 subcellular localization? Specifically, does NEDD4 translocate into nucleus or bind to chromatin under high glucose condition?

We thank for the insightful suggestion by reviewer #1. As NEDD4 Y43/585 is shown to be phosphorylated by Src upon FGF or EGF treatment¹, we first determined whether glucose could also induce the activation of Src. However, Src is not activated by glucose as indicated by its active site phosphorylation (Fig. s12c). Previous study also showed that tyrosine kinase Yes, the closest member of Src in the Src-family-kinases, is activated by glucose². Consistently, add-back glucose induced Yes phosphorylation, but not Src and Fyn (Fig. s12c). We then treated cells with Src-family-kinase inhibitor PP2, which could inhibit the kinase activity of Yes, Src, Fyn, etc. We found that glucose induced NEDD4 tyrosine phosphorylation, H3 ubiquitination and acetylation are all inhibited by PP2 (Fig. s12c, d), suggesting that SFK activation by glucose is involved in the glucose induced NEDD4 activation and may activate NEDD4 under oncogenic condition. In the future, it is our goal to study whether Yes is a direct kinase for NEDD4 and how it is involved in the cellular responses to the changes in glucose level.

We found that glucose treatment does not change NEDD4 subcellular localization and NEDD4 is presented in cytoplasm, nucleoplasm and chromatin under both glucose deprivation and high glucose condition (fig. s2c).

Reviewer #2 (Remarks to the Author):

Summary of key results:

Zhang et al report ubiquitination at histone H3 residues lysine (K) 23, 36 and 37 by E3 ligase NEDD4, and regulation of this novel histone modification by glucose. They further demonstrate that this histone H3 ubiquitination recruits Gcn5 acetyltransferase for H3K9 acetylation during transcription. Providing insight into the functions for NEDD4-mediated H3 K23/36/37 ubiquitination, the authors demonstrate a role for NEDD4 in regulating IL1 α , IL1 β and GCLM gene expression linked to tumorigenesis.

Originality and interest

While histones H2A and H2B ubiquitination are extensively studied, occurrence and functional contributions of ubiquitination at other histones remains poorly explored. Therefore, this manuscript reports a very intriguing set of findings related to novel ubiquitination modifications at histone H3. The authors have done a commendable job of characterizing the novel marks and provided functional assays linking the marks to a biological process. However, several important concerns need to be addressed prior to acceptance for publication.

Data & methodology: validity of approach, quality of data, quality of presentation

Appropriate use of statistics and treatment of uncertainties

Conclusions: robustness, validity, reliability

Suggested improvements: experiments, data for possible revision

We thank the reviewer #2 for recognizing the importance of our findings and robustness of our assays. We highly appreciate your constructive suggestions to improve our manuscript.

Major Concerns

1) NEDD4 is a well-known cytosolic protein, but the authors are reporting novel nuclear functions for this protein. Therefore, additional data are needed to convincingly demonstrate that NEDD4 localizes to chromatin to perform H3 K23/36/37 ubiquitination. Moreover, is the H3 K23/36/37 ubiquitination nuclear or cytosolic?

As suggested, we found that NEDD4 is presented on the chromatin before and after glucose treatment (fig. s2c). H3 ubiquitination could be detected on chromatin fractions in multiple data sets (fig. s1c, s3i-k, s7d). In particular, the ubiquitination on chromatin is reduced upon K23/36/37R mutation in the fig. s3k. Also, we could not detect the signal for H3 or H3 ubiquitination in the cytoplasm, suggesting that H3 K23/36/37 ubiquitination occurs in nucleus.

Does it occur on chromatin and co-localize with H3K9 acetylation? While authors have provided genome-wide data for glucose-regulated H3K9 acetylation, occupancy data for H3K23/36/37 ubiquitination to implicate these marks as a bona fide regulatory event during transcription is clearly absent. While the authors might argue the non-availability of antibodies recognizing H3 K23/36/37 ubiquitination for chromatin immunoprecipitation, it is certainly possible to determine the genome-wide occupancy of ubiquitinated form of a chromatin protein using target site(s) mutant and mono/poly-ubiquitin-specific antibody (FK2 - already used in this manuscript).

As suggested, we have now performed ChIP-seq using anti-Ub (FK2) antibody. By analyzing the pool of genes with glucose inducible H3K9ac at TSS (~2000 genes), we found that the occupancy of ubiquitinated proteins (as pulled down by FK2 antibody) is enriched at TSS (Fig. s4d), just like H3K9ac. We also found that the CHIP-seq signal is stronger in WT H3.3 cells in comparison with H3 K23/36/37R restored cells (Fig. s4d), suggesting that such ChIP-seq signal is partially related to H3 ubiquitination at K23/36/37. Together, these data revealed that H3 ubiquitination may co-localize with H3K9ac at TSS on chromatin.

2) While it is commendable that the authors acknowledge the possibility that K23/36/37 ubiquitination could also occur on canonical histones H3.1 and H3.2, they do not show any test or evidence for the presence or absence of histone H3.1 or H3.2 K23/36/37 ubiquitination, even though the manuscript title implies all H3 ubiquitination (and not just H3.3 ubiquitination) playing a role in regulating H3 acetylation and tumorigenesis. It is important for the authors to test K23/36/37 ubiquitination in histones H3.1 and H3.2 and their regulation, if any, by NEDD4 and glucose. Since H3.3 is the transcription-coupled histone H3 variant, experiments testing K23/36/37 ubiquitination in histones H3.1 and H3.2 could provide the additional compelling evidence to strongly link ubiquitination of K23/36/37 to gene transcription.

As suggested, we have now compared the ubiquitination levels between H3.1/H3.2 and H3.3 under the glucose supplemented condition. We found that the ubiquitination mainly occurs on H3.3 (either endogenous or exogenous) (Fig. s3i, j), indicating that H3.3 is the major substrate for H3 ubiquitination. Together with the fact that H3 acetylation (K9/K14) is mainly on H3.3 (Fig. 2b), these data suggest that glucose and NEDD4 mediated regulation of H3 ubiquitination and acetylation should mainly occur on H3.3 and may be additional evidence to support their role in gene transcription.

Other concerns:

1) The authors state "E3 ubiquitin ligase NEDD4 ubiquitinates histone H3 on previously unstudied lysine (K) 23/36/37 residues". While it is true that histone H3K37 residue is poorly studied, H3K23 and H3K36 are well-studied histone residues in multiple eukaryotes in the context of H3K23 acetylation and H3K36 methylation and well connected to gene transcription. Therefore, saying H3 K23 and K36 residues are unstudied is very misleading.

We agree with the reviewer and apologize for the inaccurate description. We understand that the K23 and K36 sites are well-studied for other modifications, but our intention was to say that the H3 ubiquitination on K23/36/37 is not well-studied. We have

now removed ‘previously unstudied’ from our manuscript to avoid misunderstanding.

2) While glucose add-back following starvation demonstrates its role in H3 ubiquitination, an experiment including the non-hydrolysable 2-deoxy-glucose could be included to test whether it is glucose or its breakdown that acts as an activator of H3 ubiquitination.

As suggested, we found that 2-DG add-back could not induce the H3 ubiquitination (Rebuttal fig. 1), suggesting that glucose in its intact form may not be sufficient to activate H3 ubiquitination and its breakdown metabolite(s) may be necessary for optimal H3 ubiquitination. However, it should be noted that the structure of 2-DG may still be slightly different from glucose and may not be sufficient to prove that break down of glucose is required to trigger H3 ubiquitination. It is also our future goal to study which metabolite is the key activator in this signaling pathway.

3) While the authors have tested H3 ubiquitination under multiple conditions, how H3 ubiquitination is regulated during various stages of cell cycle is missing. In general, upon reporting a novel histone modification, its establishment and regulation are usually tested in a cell synchronization-release experiment. Also, given that the authors are linking H3 ubiquitination to gene transcription, does inhibiting transcription (using amanitin, DRB, flavipiridol, etc.) adversely affect H3 ubiquitination levels?

As suggested, we performed cell cycle synchronization (Thymidine/nocodazole block and release) and tested H3 ubiquitination in the in vivo ubiquitination assay. We found that H3 ubiquitination is not regulated by cell cycle (Fig. s1d).

As suggested, we treated cells with amanitin and DRB and found that H3 ubiquitination is inhibited by these two treatments (Rebuttal fig. 2).

4) Regulation of H3 ubiquitination by NEDD4 is convincing. Therefore, on a minor note, the human genome has >600 putative E3 ligases. The authors state 'By screening a panel of ubiquitin E3 ligases', but it is not clear what rationale was employed to select these candidate E3 ligases.

Our original goal is to find an E3 ligase to access to the functional role of this newly discovered glucose induced H3 ubiquitination. Using the panel of E3 ligases available in our laboratory (now indicated in the text), we fortunately identified NEDD4 is an E3 ligase for H3. We then immediately utilized NEDD4 as a model to continue our functional study without further performing a genome wide screening. Although we cannot rule out the possibility that there are other E3 ligases involved in H3 ubiquitination, this study was to first report that glucose could trigger H3 ubiquitination through NEDD4 and its function in transcription. Screening all potential E3 ligases for H3 is our next goal and may help comprehensively understand the regulation of H3 ubiquitination under various cellular conditions.

5) When the authors examine ubiquitination levels in vivo, they claim it to be an "endogenous ubiquitination assay". This is very misleading.

As suggested, we have removed this term in the revised manuscript.

6) Figure 2 e-f, it is not clear if these two panels represent two independent experiments? Also, please edit to read, as 'H3 K23/36/37R

is defective in H3 K9/K14 acetylation'.

As suggested, we have edited the text. Indeed, these two panels are from the same samples, but ran SDS-PAGE separately. For each of the independent repeat, we ran separately for H3K9/K14ac (Fig. s3n, o).

7) The authors claim, "glucose-induced H3 ubiquitination by NEDD4 selectively regulates H3 acetylation" on page 10 and also in the title. However, the manuscript revolves primarily around H3K9 acetylation. In Figure 2, multiple histone acetylation (ac) marks in addition to H3K9ac are adversely affected upon knockdown of NEDD4. This data suggests that a sub-set of histone H3 acetylation (but not all H3 acetylation) is regulated by NEDD4. The authors do not provide any reasoning for why they chose to restrict their study to H3K9ac, especially since NEDD4 knockdown in fact completely abolishes H3K14ac!

We agree with the reviewer #2 that the H3 acetylation at certain lysine sites are not changed upon NEDD4 knockdown and glucose treatment. As suggested and as we did in most cases, we have now modified the text to 'NEDD4 selectively regulates H3 acetylation at specific lysine sites, including K9 and K14' to avoid misunderstanding. Due to length limitation of the title, it is hard to clearly explain which sites are affected and which sites are not.

The reasons why we primarily study H3K9ac, especially in ChIP-seq, are two folds: 1) the antibody for H3K9ac is much better/stronger than H3K14ac in both Western blot and ChIP experiments. 2) The fold induction by glucose is higher for H3K9ac than H3 acetylation at other lysine sites, including K14 (Fig. S3a, b). Although we did not include H3k14ac in some of the assays due to the technical reason, we speculate that H3K14ac should still follow the trend of H3K9ac upon the changes in glucose and/or NEDD4 as shown in Western blot assay for multiple cell types.

8) In Figure 2, panels a - b, why is the signal for H3K36ac so variable?

The conditions in 2a and 2b are different. While Fig. 2b shows the basal H3 acetylation under the glucose proficient condition, Fig. 2a represents the recovery of H3 acetylation by adding-back glucose in glucose-depleted condition. These data suggest that glucose starvation completely wiped out H3K36ac, but unlike H3 acetylation at other lysine sites, 3 hours of glucose treatment did not seem to give enough time to recover H3K36ac to the normal level from glucose starvation.

9) In Figure 4d, expression of many histone genes are reduced following knockdown of NEDD4. A closer inspection of various H3 transcript levels and H3.1, H3.2 or H3.3 protein levels (using a more dynamic range) are needed to eliminate the possibility that the reduction or loss of ubiquitination is a consequence of reduced transcript/protein levels.

In fact, each histone is expressed from multiple genes/multiple transcripts. A change in the transcription of one such gene may not affect the histone protein level, which is a more functional indicator. As suggested, we have checked the protein level of H3.1/H3.2 and H3.3 under low exposure and found that their levels are not reduced upon NEDD4 knockdown (Fig. s3h). Thus, the reduction of H3 ubiquitination in NEDD4 deficient cells is not due to the uneven H3 protein levels.

10) In Figure 5i, the authors show IP/pull-down assays to demonstrate the interaction between GCN5 and ubiquitinated H3. A reciprocal complementary test of H3ubi interaction with Flag-Gcn5 could also be included. As a direct test of their recruitment model, the authors could test the binding of GCN5 and/or GCN5-containing complex interaction to NEDD4-ubiquitinated nucleosomes in vitro, and include nucleosomes with H3 K23/36/37R as a control to further demonstrate the specificity for the interaction.

In our manuscript, we utilized his-tag pull-down (of his-Ub) in the denaturing guanidine-HCl based buffer to verify the existence of ubiquitinated H3 from Flag-GCN5 immuno-complex. By using such strong detergent, we can exclude the scenario that the pull-down of ubiquitinated-H3 from Flag-GCN5 immuno-complex is the result of indirect pull-down of other ubiquitinated interacting proteins. Thus, there is not a good reciprocal assay design for this experiment, as to pull-down ubiquitinated-H3 first in the guanidine-HCl based buffer will disrupt any protein-protein interaction. And using milder buffer will lead to indirect interaction and false positive result.

As suggested, we included a more direct test suggested by the reviewer #2. We found that GCN5-containing complex purified from cells preferentially binds to in vitro ubiquitinated H3 comparing to un-ubiquitinated H3 (Fig. s7c). H3 K24R was used as a ubiquitination deficient control.

Appropriate use of statistics and treatment of uncertainties
No concerns

References: appropriate credit to previous work?
Yes

Clarity and context: lucidity of abstract/summary, appropriateness of abstract, introduction and conclusions
Please refer to the comments above to improve the clarity and provide more data to arrive to support the conclusions of the manuscript.

Reviewer #3 (Remarks to the Author):

Zhang et al present a study investigating epigenetic regulation in cancer through NEDD4-dependent H3 ubiquitination. The authors find that E3 ubiquitin ligase NEDD4 is induced by glucose to ubiquitinate histone H3 on previously unstudied lysine sites K23, K36, and K37. Using

ChIP-seq, the authors discover that NEDD4 regulates glucose-induced H3K9 acetylation at the transcription starting site and enhancer regions. Further, authors claim that NEDD-mediated H3 ubiquitination regulate tumor sphere formation by inducing transcription of target genes IL1 α , IL1 β , and GCLM.

We are grateful for your constructive suggestions allowing us to improve the quality of our manuscript.

Major comments:

1. In Figure 2a, the authors investigate glucose-induced H3 acetylation at H3 ubiquitination sites identified in large-scale quantitative proteomics studies shown in Figure 1a. However, not all sites are included: K37, K79 and K122 are missing. Why were these sites not investigated?

We would like to point out that we are not selecting H3 acetylation sites through H3 ubiquitination sites in fig. 1a. Indeed, we designed the experiments to examine the effect of NEDD4 on all well-studied H3 acetylation sites in fig. 2a. Fig. 1a illustrates potential H3 ubiquitination sites as discovered by mass spectrum, which is not related to the assay for H3 acetylation at various sites in fig. 2a. In addition, since K37, K79 and K122 are not well-defined sites for H3 acetylation and their functions are also not well characterized, we therefore do not examine these sites.

2. Statistical analysis is missing for Figures 4e, 4f, 4h, 4i, 4j, 5a, 5b, 5d, 7c, 7d, and 7e. Therefore, it is not clear whether the observed differences are statistically significant.

They are all statistically significant and we have now included the statistical test in these figures.

3. To confirm the potential role of NEDD4 in cancer, the authors use Hep3B cells for in vitro tumor sphere formation. Why were Hep3B cells used? At least two more human cancer cell lines should be used to

convincingly demonstrate that NEDD4 plays a role in tumorigenesis.

While glucose starvation drastically inhibited H3 acetylation in the Hep3B cells (like all other cell types), it is more resistant to the glucose starvation induced cell death than many other cancer cell lines, providing a good and easy-to-handle model to study the effect of glucose add-back.

As suggested, we have also included more data for MDA-231 and Du145 cell lines (we already had some data for those cell lines in our previous manuscript) to better demonstrate the role of NEDD4 in H3 acetylation and tumorigenesis (Fig. s1h, s3d, e, s5b, s9b, c, d, e).

4. The authors do not specify which cancer cells were used for the in vivo tumor engraftment assay. How were these cells selected?

We apologize that we forgot to include the cell line information in the previous manuscript and have now indicated in the figure legends. We used Du145 in the in vivo tumor engraftment assay, since our lab has confirmed that Aldh+ is a robust cancer initiating cell marker to sort Du145 cells for in vivo study. Unlike in vitro sphere formation, we have difficulty in getting tumor engraftment in vivo using Aldh+ as a marker for Hep3B.

Minor comments:

1. Aldh should be defined in the manuscript.

We have now defined Aldh+ as high aldehyde dehydrogenase activity when it first appeared in the text.

2. There are several typos throughout the manuscript. Some examples include:

a. "...(HATs) that are often resides..."

b. "...which is also differ from..."

c. "S.E.M." Only one period should be used at the end of each

sentence.

As suggested, we have now corrected those grammar errors in the revised manuscript.

Reviewer #4 (Remarks to the Author):

The manuscript by Zhang et al. entitled “H3 ubiquitination by NEDD4 regulates H3 acetylation and tumorigenesis” describes the regulation of H3 acetylation by ubiquitination during glucose stimulation. The authors demonstrate glucose-induced ubiquitination of H3 by NEDD4 and involvement of tyrosines 43 and 585 phosphorylation during the process (Figure 1). They further show that the NEDD4-dependent ubiquitination of the histone H3 variant H3.3 at lysines 23, 36 and 37 is critically required for the acetylation of H3K9, K14, K27 and K56 (Figure 2). Knocking down of NEDD4 compromises glucose-induced H3 acetylation at TSS and enhancer regions (Figure 3). Histone H3 acetylation induced by glucose at a large fraction of promoters and enhancers is dependent on the presence of activity of NEDD4 (Figure 4). The authors show that the NEDD4-mediated ubiquitination of H3 recruits GCN5, which is responsible for the glucose-induced H3 acetylation (Figure 5). They further demonstrate the involvement of the NEDD4 pathway in controlling tumor cell growth both in vitro and in a mouse model (Figure 6).

Comments: Histones are known to be heavily modified by acetylation, methylation and ubiquitination etc. Cross-talks between different modifications are important for regulation of transcription. In this study, the authors characterized several H3 ubiquitinations previously revealed by proteomics studies. They identified the enzyme NEDD4 that is responsible for the ubiquitination and its function in facilitating H3 acetylation during glucose stimulation and further demonstrate its in vivo function controlling tumor cell growth. This is an interesting story, which provides novel information. The experiments are well designed and the data are clean. The manuscript is well written. I feel the manuscript should be accepted for publication with a few minor modifications.

We thank the reviewer #4 for highly recognizing our work and indicating that our study is interesting and suitable for publication.

1. Figure 5c: the procedure discovering the potential regulators is not clearly described.

We utilized iRegulon software to compare the NEDD4 microarray data to all published ChIP-seq tracks for various transcription factors. If the pattern of certain ChIP-seq track is similar to the NEDD4 microarray data, the protein pulled down in such ChIP-seq experiment could be the potential regulator of NEDD4 mediated transcription pattern. All potential regulators are ranked by statistical methods. Since this method is not developed by us, we only briefly described the principal of this prediction and cited the original paper for iRegulon in our manuscript for detailed information.

2. Figure S3l: Is this experiment done once only? No error bars are included.

We have now performed 3 times of such experiments, and error bar was included.

3. Page 10. Line 7 from top: the reference for H3K9 acetylation at TSS is not the most appropriate one. The following papers should be cited: (Genes Dev. 2005 Mar 1;19(5):542-52.; Nat Genet. 2008 Jul;40(7):897-903.)

As suggested, we have now cited more appropriate references as indicated by reviewer #4.

Rebuttal References

- 1 Persaud, A. *et al.* Tyrosine phosphorylation of NEDD4 activates its ubiquitin ligase activity. *Science signaling* **7**, ra95, doi:10.1126/scisignal.2005290 (2014).
- 2 Yoder, S. M., Dineen, S. L., Wang, Z. & Thurmond, D. C. YES, a Src family kinase, is a proximal glucose-specific activator of cell division cycle control protein 42 (Cdc42) in pancreatic islet beta cells. *The Journal of biological chemistry* **289**, 11476-11487, doi:10.1074/jbc.M114.559328 (2014).

REVIEWERS' COMMENTS:

Reviewer #1 (Remarks to the Author):

The authors have addressed the questions in a satisfactory manner. The manuscript is suitable for publication.

Typo: Line 192, S2j should be S3j.

Reviewer #2 (Remarks to the Author):

The authors have thoroughly addressed all my concerns with additional experimental data and the manuscript quality has been substantially increased.

The manuscript is now ready for publication.

Reviewer #3 (Remarks to the Author):

Points raised in the previous round of review have been satisfactorily addressed.

Reviewer #4 (Remarks to the Author):

The authors satisfactorily addressed my and other reviewers' questions. I think the manuscript is ready for publication.